# LIRE: LISTWISE REWARD ENHANCEMENT FOR PREFERENCE ALIGNMENT

## ABSTRACT

Recently, tremendous strides have been made in the domain of Natural Language Generation (NLG) due to the vast advances in Large Language Models (LLMs). However, often trained on large-scale unsupervised data, LLMs may generate toxic or unhelpful content for lack of human supervision. Leveraging reinforcement learning with human feedback (RLHF) turns out a good remedy for this problem and has been prevalent among researchers. However, RLHF is notoriously unstable and hyperparameter-sensitive, which hinders an all-compassing and sustainable LLM system. For the above reason, we propose a new approach: LIRE, which stands for *Listwise Reward Enhancement for Preference Alignment*, to optimize rewards through a listwise paradigm. We directly incorporate the rewards of multiple candidates into the listwise loss and optimize against it in a compact and effective framework, without explicit modeling of the Bradley-Terry model. Furthermore, we propose a self-enhancement algorithm to progressively optimize the reward through iterative training. Our work also entails extensive experiments to demonstrate the stability and consistency of the model performance without heavy hyperparameter tuning, while still surpassing the state-of-the-art methods in preference alignment tasks.

## 1 INTRODUCTION

While a growing plethora of large language models (LLMs) have exhibited incredible performance in a broadening scope of tasks and applications such as summarization, machine translation, and dialog generation Nakano et al. (2021); Stiennon et al. (2020); Brown et al. (2020); Zhao et al. (2023a), they can still output contents that are harmful, biased or simply do not agree with standard human perception Mathur et al. (2020); Fernandes et al. (2023). This is an inherent problem existing in the extensive data sources during model training Ouyang et al. (2022); Bai et al. (2022); Song et al. (2023), and can be alleviated by incorporating certain restrictions or limitations to align the output generation towards human desires and specifications Ngo (2022); Kenton et al. (2021). Existing methods focus on employing reinforcement learning from human feedback (RLHF) to fine-tune the pre-trained LLMs Christiano et al. (2017); Stiennon et al. (2020); Ouyang et al. (2022); Xue et al. (2023), which concept was originally introduced in the field of robotics and Atari games Christiano et al. (2017); Ibarz et al. (2018). RLHF in LLM introduces a paradigm that involves leveraging supervised fine-tuning (SFT) on the initial models, fitting the reward model to human preferences, and then using Reinforcement Learning (RL) algorithms such as Proximal Policy Optimization (PPO) Schulman et al. (2017) to optimize a policy that doesn't drift overly far from the original model Rafailov et al. (2023). Such methods successfully incorporate human preferences into data training and achieve satisfying results to a large extent.

However, PPO is trained in a pointwise manner and optimizes at every single step based on the rewards, penalizing fragments within a segment equally and disregarding the truly informative parts. Alternatively, pairwise ranking leverages a comparison between a positive and a negative sample to incorporate context information. Methods such as DPO Rafailov et al. (2023), PRO Song et al. (2023), and RRHF Yuan et al. (2023) all leverage a pairwise comparison model to optimize the rewards. Nevertheless, the performance of pairwise ranking is heavily dependent on the quality of the sample pairs, and trivial negatives may yield suboptimal results. Moreover, if given a large candidate pool, performing pairwise comparisons among multiple samples entails a significant computation complexity.

For the above reasons, we propose a listwise optimization approach: *Listwise Reward Enhancement for Preference Alignment* (LIRE). Instead of employing the Bradeley-Terry model Bradley & Terry (1952) or Plackett-Luce models Plackett (1975) to rank the candidates, we take a listwise approach by modeling the response probability distribution under the general policy gradient framework, with reward scores implicitly weighing samples differently during loss calculation. Essentially, LIRE does not rely on an ordinal ranking, instead, the ranking information is implicitly given by the reward scores. This is different from the top-k probability defined in ListNet Cao et al. (2007), which gives a permutation probability distribution that relies on the position of a response in the permutation. LIRE considers multiple responses simultaneously at each iteration and is therefore free from hard mining techniques to eliminate the influence of trivial negatives.

We give the training pipeline of the proposed LIRE in Figure 1. The overarching concept is as follows: we first construct the candidate pool by gathering responses $A$ for queries $Q$ from different initial policies $\pi_{\theta_{init}}$. A popular approach to gathering data is to utilize LLM generations with various decoding strategies. Note that human preference data is also a kind of sampling data and constitutes our reservoir of candidates. After the responses are gathered, we have the environment to provide rewards $R$ and then leverage a listwise optimization approach. The updated model $\pi_\theta$ is re-initialized as the sampling policy and generates fresh responses that substitute the prior ones within the candidate pool. Through iterative training, the model progressively enhances the ability for preference alignment.

Extensive experiments of the state-of-the-art methods are fairly conducted on multiple benchmarks of dialogue generation and summarization tasks. The results show that the proposed LIRE achieves superior and consistent performance in all the experiments, exhibiting more noticeable gains as we increase the size of the candidate pool.

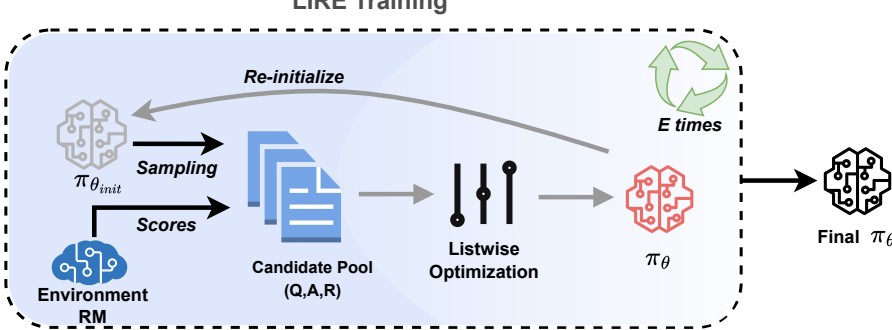

Figure 1. Training pipeline of the proposed LIRE framework. The candidate pool is initially constructed by gathering responses $A$ with different policies $\pi_{\theta_{init}}$ and rewards $R$ from the environment (Reward Model) before they are optimized in a listwise manner. The updated model $\pi_\theta$ is then re-initialized as the sampling policy and generates fresh responses that substitute the prior ones within the candidate pool. Through iterative training, the model progressively enhances the ability for preference alignment.

## 2 RELATED WORK

Leveraging human feedback to improve model generation ability toward human desire renders it imperative given the quickly growing family of LLMs. Directly leveraging human feedback to optimize models generally requires an "optimizable" formulation of the feedback Fernandes et al. (2023). However, it is expensive and impractical to generate sufficient human feedback for LLM training in general cases, whether numerical, ranking-based, or even natural language-based. As an alternative, one line of work relies on models to produce feedback that approximates human perception Stiennon et al. (2020); Ouyang et al. (2022); Askell et al. (2021).

Given enough feedback (preference data), RLHF has been extensively employed to optimize an LLM with various training objectives using a unified approach. SFT is an alternative approach that involves maximizing the likelihood of the top-1 candidate directly Zhou et al. (2023); Thoppilan et al. (2022). Both methods can be used in tandem as demonstrated in Ouyang et al. (2022), where InstructGPT is proposed to steer model generation better towards human instruction and desire. In the typical setting of RLHF, the model is first fine-tuned with the preference datasets, followed by

a reward modeling procedure that gives scores to model output. Finally, RL policies are utilized to maximize the overall reward. This is an online procedure that requires multiple sampling from the updated policy and scoring during training, thus suffering complex training and high computation costs Gulcehre et al. (2023). Many methods have aimed to improve efficiency as well as performance for preference alignment over online RL policies such as PPO. DPO Rafailov et al. (2023) reformulates the constrained reward maximization problem as a direct policy optimization problem by correctly classifying the preference data, which proves to be performant and computationally lightweight. SLiC-HF Zhao et al. (2023b) utilizes the rank calibration loss and cross-entropy regularization loss to learn pairwise human feedback. Other approaches employ ranking-based methods to align preferences, which naturally extend beyond binary-format preference data. RRHF Yuan et al. (2023) learns to align scores of sampled responses with human preferences through pairwise ranking loss among multiple responses. PRO Song et al. (2023) iteratively contrasts the likelihood of the best response against the remaining responses on a rolling basis, using an extended pairwise Bradley-Terry comparison model. These methods consider not only the positive-labeled responses, as in the typical SFT loss, but also negative samples. Another line of work directly utilizes reward scores from reward models for filtering purposes to improve model generation. *ReST* Gulcehre et al. (2023) introduces two loops and frames the alignment problem as a growing batch RL problem. The outer loop is a Grow step that iteratively augments the training dataset, and the inner loop is an Improve step that involves filtering the generated data and fine-tuning a model on the filtered dataset with offline RL algorithms. Concurrent to this work, RAFT Dong et al. (2023) subsequently selects the $1/k$ percent of samples with the highest reward as the training samples and then fine-tune the model on this filtered dataset.

While the above methods all bring improvement to better aligning model output with human preferences, we believe more research and effort should be devoted to this research topic. To the best of our knowledge, reward scores so far have not been explicitly integrated into the training objective, mainly limited to a filter function at most for data selection in offline settings such as in Dong et al. (2023); Gulcehre et al. (2023). Besides, the idea of listwise optimization has not yet been fully studied in this domain. In this paper, we introduce a framework that directly optimizes the expectation of rewards in a listwise fashion, and makes the model more "steerable".

## 3 PRELIMINARIES

In this section, we illustrate the motivation for the LIRE framework and the related preliminaries. To start with, we give the optimization objective in the common RLHF settings Ouyang et al. (2022); Stiennon et al. (2020); Ziegler et al. (2019):

$$\max_{\pi_\theta} \mathbb{E}_{\mathbf{x}\sim\mathcal{D}, \mathbf{y}\sim\pi_\theta(\mathbf{y}|\mathbf{x})} \bigg( r_\phi(\mathbf{x}, \mathbf{y}) \bigg) - \beta\mathbb{D}_{\mathrm{KL}}\bigg( \pi_\theta(\mathbf{y}|\mathbf{x})||\pi_{\mathrm{ref}}(\mathbf{y}|\mathbf{x}) \bigg), \qquad (1)$$

where $r_\phi$ is the well-trained reward function, and $\pi_{\mathrm{ref}}$ and $\pi_\theta$ are the reference policy and the LM policy, respectively. Rafailov et al. (2023) gives the optimal policy of the above KL-constrained objective and further derives this optimal policy under the famous Bradley-Terry model to model the preference. These methods directly or implicitly stem from Equation 1 and are thus always heavily dependent on the KL constraint.

In view of the above reasons, we move one step back and start with the original policy gradient methods in RL. The general and coarser expression for the optimization objective in RLHF can be formulated as:

$$J(\theta) = \mathbb{E}_{\mathbf{x}\sim\mathcal{D}, \mathbf{y}\sim\pi_\theta(\mathbf{y}|\mathbf{x})} R(\mathbf{x}, \mathbf{y}) = \sum_{\mathbf{y}, \mathbf{x}} P_{\pi_\theta}(\mathbf{y}|\mathbf{x}) R(\mathbf{x}, \mathbf{y}), \qquad (2)$$

where $P_{\pi_\theta}$ is the probability distribution of the trajectory under some policy $\pi_\theta$, and $R(\mathbf{x}, \mathbf{y})$ is the reward model that provides reward signals during training. The ultimate goal of policy gradient methods is to maximize the rewards of the trajectories under the policy $\pi_\theta$. Since this is an on-policy process, the training data has to be sampled iteratively as policy $\pi_\theta$ updates. PPO is a popular method that turns this on-policy learning into an off-policy process, by resorting to importance sampling as well as the KL penalty to approximate the true distribution of the unknown $P_{\pi_\theta}(\mathbf{y}|\mathbf{x})$ Schulman et al. (2017). In this paper, we propose an alternative to approximate $P_{\pi_\theta}(\mathbf{y}|\mathbf{x})$ with sampled responses and $R(\mathbf{x}, \mathbf{y})$ with the reward scores. Specifically, our method initially models the probability distribution with the generated responses from LLMs and scores the responses using

well-trained reward models. Subsequently, it optimizes the expectation of the final rewards in a listwise manner.

## 4 METHODOLOGY

### 4.1 LIRE: LISTWISE REWARD ENHANCEMENT FOR PREFERENCE ALIGNMENT

In this section, we reformulate the preference alignment problem and introduce a listwise softmax loss in our LIRE framework. As illustrated in Figure 1, our framework comprises two main components: offline data generation and online model training. In the offline phase, we assume a set of queries $\mathbf{Q} = \{\mathbf{x}^{(1)}, \mathbf{x}^{(2)}, \cdots, \mathbf{x}^{(N)}\}$ is given, and each query is associated with a list of offline responses $\mathbf{A}^{(i)} = \{\mathbf{y}_1^{(i)}, \cdots, \mathbf{y}_m^{(i)}\}, i \in \{1, \cdots, N\}$. Furthermore, each response $\mathbf{y}_j^{(i)}$ for query $\mathbf{x}^{(i)}$ is paired with a score $R(\mathbf{x}^{(i)}, \mathbf{y}_j^{(i)})$ by some reward model RM.

During training, we aim to learn a language model parameterized by $\theta$, which generates responses with better alignment with human preferences. First, we define a set of token prediction probabilities conditioned on $\mathbf{x}^{(i)}$ as $\mathbf{P}_{\pi_\theta}(\mathbf{y}_{j,k}^{(i)}|\mathbf{x}^{(i)}) \in \mathbb{R}^{L \times V}$, where $L$ is the sequence length and $V$ the vocabulary size. The probability of the sentence $\mathbf{y}_j^{(i)}$ with $K$ tokens are formulated as:

$$\pi_\theta(\mathbf{y}_j^{(i)}|\mathbf{x}^{(i)}) = \prod_{k=1}^{K} \mathbf{P}_{\pi_\theta}(\mathbf{y}_{j,k}^{(i)}|\mathbf{x}^{(i)}, \mathbf{y}_{j,<k}^{(i)}). \tag{3}$$

Next, the probability of the response distribution against response set $\mathbf{A}^{(i)}$ is calculated as:

$$P_{\pi_\theta}(\mathbf{y}^{(i)}|\mathbf{x}^{(i)}, \mathbf{A}^{(i)}) = \frac{\exp(\frac{1}{T} \log \pi_\theta(\mathbf{y}^{(i)}|\mathbf{x}^{(i)}))}{\sum_{j=1}^{m} \exp(\frac{1}{T} \log \pi_\theta(\mathbf{y}_j^{(i)}|\mathbf{x}^{(i)}))}, \tag{4}$$

where $T$ is a temperature parameter to control the smoothness of the probability distribution.

So far we have given an approximation of the $P_{\pi_\theta}$ in Equation (2), we next derive the listwise loss of our LIRE objective. The general idea is that the quantized scores provide more specific and direct guidance to the model during training, compared to solely based on cardinal ranking numbers. Formally, the loss is calculated as:

$$J(\theta) = -\sum_{i=1}^{N} \mathbb{E}_{\mathbf{y}^{(i)} \sim \pi_\theta(\mathbf{y}^{(i)}|\mathbf{x}^{(i)})} R(\mathbf{x}^{(i)}, \mathbf{y}^{(i)})$$
$$= -\sum_{i=1}^{N} \sum_{j=1}^{m} P_{\pi_\theta}(\mathbf{y}_j^{(i)}|\mathbf{x}^{(i)}, \mathbf{A}^{(i)}) R(\mathbf{x}^{(i)}, \mathbf{y}_j^{(i)}). \tag{5}$$

In practice, we apply softmax to the reward scores of a single query $R(\mathbf{x}^{(i)}, \mathbf{y}^{(i)})$ due to its property of translation invariance. By doing so we mitigate the influence of different reward scales and maintain stable training parameter settings. To this end, we successfully derived the listwise loss of our LIRE objective. The sophisticated modeling of pairwise comparison among multiple responses has been safely circumvented and the objective in Equation (5) nicely resonates with our initial goal in Equation (2). To develop a general perception of what the model actually learns through the process, we next illustrate the derivative of $J(\theta)$ with regard to model parameters $\theta$. We also give a detailed derivation process in Appendix A.1.

$$\nabla J(\theta) = -\frac{1}{T} \sum_{i=1}^{N} \mathbb{E}_{\mathbf{y}^{(i)} \sim \pi_\theta(\mathbf{y}^{(i)}|\mathbf{x}^{(i)})} \left[ \frac{\nabla P_{\pi_\theta}(\mathbf{y}^{(i)}|\mathbf{x}^{(i)}, \mathbf{A}^{(i)})}{P_{\pi_\theta}(\mathbf{y}^{(i)}|\mathbf{x}^{(i)}, \mathbf{A}^{(i)})} \right.$$
$$\left. \times \left( R(\mathbf{x}^{(i)}, \mathbf{y}^{(i)}) - \mathbb{E}_{(\mathbf{y}'^{(i)} \sim \pi_\theta(\mathbf{y}^{(i)}|\mathbf{x}^{(i)}))} R(\mathbf{x}^{(i)}, \mathbf{y}'^{(i)}) \right) \right]. \tag{6}$$

It shows that $\frac{\nabla P_{\pi_\theta}(\mathbf{y}^{(i)}|\mathbf{x}^{(i)}, \mathbf{A}^{(i)})}{P_{\pi_\theta}(\mathbf{y}^{(i)}|\mathbf{x}^{(i)}, \mathbf{A}^{(i)})}$ is the normalized gradient of model predictions, multiplied by a demeaned reward score. These demeaned rewards act as a weighting mechanism that encourages responses with higher scores while depressing those with lower rewards.

**Relation with pairwise losses and DPO.** When the number of candidate responses descends to 2, this listwise loss degenerates into a pairwise loss. Specifically, we rewrite Equation (6) into a pairwise formulation under 2 responses (omitting $\mathbf{A}^{(i)}$ for clarity):

$$\nabla J_{\text{LIRE-2}}(\theta) = -\frac{1}{T} \sum_{i=1}^{N} \left[ P_1 \times \nabla P_{\pi_\theta}(\mathbf{y}_1^{(i)}|\mathbf{x}^{(i)}) + P_2 \times \nabla P_{\pi_\theta}(\mathbf{y}_2^{(i)}|\mathbf{x}^{(i)}) \right], \quad (7)$$

where $P_j = \frac{P_{\pi_\theta}(\mathbf{y}_j^{(i)}|\mathbf{x}^{(i)})^{(\frac{1}{T}-1)}}{\sum_m P_{\pi_\theta}(\mathbf{y}_m^{(i)}|\mathbf{x}^{(i)})^{\frac{1}{T}}} \times \delta R(\mathbf{x}^{(i)}, \mathbf{y}_j^{(i)})$, and $\delta R(\mathbf{x}^{(i)}, \mathbf{y}_j^{(i)})$ is the corresponding demeaned reward scores, $j \in \{1, 2\}$, $m = 2$. Referring to our previous definition format, we reorganized the gradient of the DPO objective in the following:

$$\nabla J_{\text{DPO}}(\pi_\theta; \pi_{\text{ref}}) = -\beta \sum_{i=1}^{N} \left[ r \times \nabla \log \pi_\theta(\mathbf{y}_1^{(i)}|\mathbf{x}^{(i)}) + (1-r) \times \nabla \log \pi_\theta(\mathbf{y}_2^{(i)}|\mathbf{x}^{(i)}) \right], \quad (8)$$

with $r$ defined by the policy $\pi_\theta$ and reference model $\pi_{\text{ref}}$. Interestingly, these two objectives resemble in that they can both be viewed as the weighted sum of gradients of two responses, with higher weights for preferred responses and lower weights for rejected ones. The difference is that in our LIRE, $P_j$ is determined by offline rewards together with the model predictions. In DPO, $r$ is determined by the differences in the rewards of two responses.

Interestingly, we can further substitute $\nabla P_{\pi_\theta}(\mathbf{y}_j^{(i)}|\mathbf{x}^{(i)})$ with $\nabla \log \pi_\theta(\mathbf{y}_j^{(i)}|\mathbf{x}^{(i)})$ through some algebra and align the derivative objectives. Subsequently, our objective in Equation (7) takes the form:

$$\nabla J_{\text{LIRE-2}}(\theta) = -\frac{1}{T^2} \sum_{i=1}^{N} \left[ \tilde{P}_1 \times \nabla \log \pi_\theta(\mathbf{y}_1^{(i)}|\mathbf{x}^{(i)}) + \tilde{P}_2 \times \nabla \log \pi_\theta(\mathbf{y}_2^{(i)}|\mathbf{x}^{(i)}) \right], \quad (9)$$

where $\tilde{P}_j = \frac{P_{\pi_\theta}(\mathbf{y}_j^{(i)}|\mathbf{x}^{(i)})^{\frac{1}{T}}(1-P_{\pi_\theta}(\mathbf{y}_j^{(i)}|\mathbf{x}^{(i)})^{\frac{1}{T}})}{\sum_m P_{\pi_\theta}(\mathbf{y}_m^{(i)}|\mathbf{x}^{(i)})^{\frac{1}{T}}} \times \delta R(\mathbf{x}^{(i)}, \mathbf{y}_j^{(i)})$. This way, the relation between LIRE and DPO becomes clearer. Please refer to Appendix A.2 for detailed derivation.

## 4.2 THE SELF-ENHANCEMENT ALGORITHM

---

**Algorithm 1:** The self-enhancement strategy for reward maximization during progressive sampling and consecutive training process. An *Evolve* step is defined as a data generation procedure with policy $\pi_\theta$, followed by subsequent *Iterate* steps of policy training with regard to objective $J(\theta)$.

---

**Input:** Input queries $\mathbf{x}$, training objective $J(\theta)$, reward model RM, number of samples per query $m$, Language Model with initial policy $\pi_{\theta_{\text{init}}}$, *Evolve* steps $E$, *Iterate* steps $I$.

1 **for** $e = 1$ *to* $E$ **do**
2      Generate dataset $D_e$: for each query $\mathbf{x}^{(i)}$, sample $m$ responses $\mathbf{A}^{(i)} \sim \pi_\theta(\mathbf{y}|\mathbf{x}^{(i)})$.
3      Score $D_e$ with the reward model RM.
4      **for** $i = 1$ *to* $I$ **do**
5          Update $\pi_\theta$ on data $D_e$ with the objective $J(\theta)$.
6      **end**
7 **end**

**Output:** The learned policy $\pi_\theta$.

---

To further boost the performance, we propose Algorithm 1 to conduct iterative data sampling and incremental policy updates. This iterative strategy is also adopted in works Gulcehre et al. (2023); Dong et al. (2023) and proves to be effective. The whole training outline are divided into two phases: Data Sampling (*Evolve*) and Policy Training (*Iterate*). We start by sampling responses from some policy $\pi_{\theta_{\text{init}}}$, and this can be pretrained LLMs or human preference, then we score the responses with some reward model RM. Afterwards, we initialize the target policy $\pi_\theta$ as the pretrained LLM and start to optimize the objective $J(\theta)$ in Equation (5). The current model again samples completions to construct a new candidate pool. One approach is to only keep new candidates with higher reward scores and discard those degraded ones, this way we can better ensure the policy is updated on a higher-quality dataset and prevent policy diverging. Specifically, $E = 1$ suggests we sample responses only once and then conduct training, without iterative sampling afterwards.

| Test Data | Eval Metric | ø | PPO | DPO | PRO | RRHF | LIRE |
|-----------|-------------|-----|-----|-----|-----|------|------|
| HH Test | PPL | **10.98** | 11.81 | 16.04 | 16.63 | 14.66 | 12.15 |
|  | RM | -0.93 | -0.96 | -0.87 | -1.02 | -0.96 | **-0.85** |

Table 2. **Comparison of LIRE and other methods on Anthropic HH Dataset**. ø refers to zero-shot results of Alpaca-7B. The best and second best results are marked with **Bold** and underlined format.

## 5 EXPERIMENTS

### 5.1 DATASETS

For performance comparison, We mainly focus on dialogue generation and summarization tasks. For dialogue, we use Anthropic's Helpful and Harmless (HH) dataset. Moreover, in order for a more diverse candidate pool, we sample responses with LLM completions due to their impressive language generation abilities. We follow Yuan et al. (2023) to sample responses from Alpaca-7B Taori et al. (2023) using diverse beam search. All the responses of a single query are scored by reward model RM. For summarization, we use the *TL;DR* Summarization dataset from Stiennon et al. (2020) and score the resulting responses by RM-SUM.

### 5.2 COMPARISON METHODS

To demonstrate the ability of the proposed LIRE, we conduct an exhaustive investigation into the state-of-the-art methods on human preference alignment tasks. PPO is implemented according to the official code from trlx. DPO Rafailov et al. (2023) optimizes the constrained reward maximization problem in PPO using a single stage of policy training, so it is essentially easier to train and achieves better performance than PPO. PRO Song et al. (2023) and RRHF Yuan et al. (2023) are two preference ranking methods that both support multiple-response ranking. We follow the default configuration settings introduced in the official codes for each method and Lora Hu et al. (2021) is applied for the concern of computation and memory limitation. We implement these methods on Alpaca-7B as the base model. More implementation details can be found in Appendix A.4.

### 5.3 COMPARE AGAINST THE STATE-OF-THE-ARTS

Firstly we conduct a thorough assessment of the methods introduced in Section 5.2 on the Human Preference HH dataset. The automatic evaluation is directed on HH test. We leverage Perplexity (PPL) using gpt2-medium and reward model RM. Since the reward score is our optimization target, we focus more on the analysis of this evaluation indicator.

As shown in Table 2, when trained with the HH dataset, LIRE achieves the best performance with regard to the average reward score, with DPO attaining the second-best reward score at the sacrifice of a much lower PPL. As for PPO, it achieves a smaller PPL, very close to the zero-shot results. Our hypothesis is that models trained in a pointwise manner focus more on a single data sample, thus giving more coherent and certain predictions based on the preceding

| vs. | ø | PPO | DPO | PRO | RRHF | HW |
|-----|-----|-----|-----|-----|------|-----|
| **HW win %** | 48 | 49 | 48 | 55 | 56 | - |
| **LIRE win %** | 61 | 50 | 54 | 62 | 60 | 56 |

Table 1. **Win rates (%) from human evaluation on a subset of Anthropic-HH**. The first row gives win rates for human-written (HW) responses versus different methods, and the second row stands for direct comparison between LIRE versus other methods. Win rates greater than or equal to 50 are marked in orange.

context. Besides, Table 1 gives human evaluation on a subset of Anthropic-HH. The first row is for human-written responses versus different methods, and the second row is for comparing LIRE against other methods directly. LIRE achieves the highest win rate, which is in line with the results of automatic metrics. More details can be found in Appendix A.7.

We also leverage the TL;DR summarization task to validate the proposed LIRE framework in Table 3. To avoid possible model hacking Skalse et al. (2022); Touvron et al. (2023) behavior or inflated reward scores due to overfitting, we additionally utilize another reward model RM-SUM* to evaluate the methods. Note that RM-SUM* and RM-SUM are two different training versions of the same model, and should have similar judgments toward the model responses. We employ RM-SUM* to investigate how the models perform under a reward criterion, which is not identical

| Test Data | Eval Metric | ø | PPO | DPO | PRO | RRHF | LIRE |
|---|---|---|---|---|---|---|---|
| | Rouge-L | 0.096 | 0.16 | 0.29 | **0.32** | 0.20 | 0.22 |
| TL;DR | RM-SUM | -1.74 | 1.16 | 2.14 | 1.49 | 1.35 | **2.76** |
| | RM-SUM* | -0.31 | 2.09 | 1.89 | 1.15 | 0.82 | **2.79** |

Table 3. **TL;DR Summarization results of different methods**. LIRE got the highest reward scores for both RM-SUM and RM-SUM*, with DPO and PPO attaining the second-highest scores, respectively.

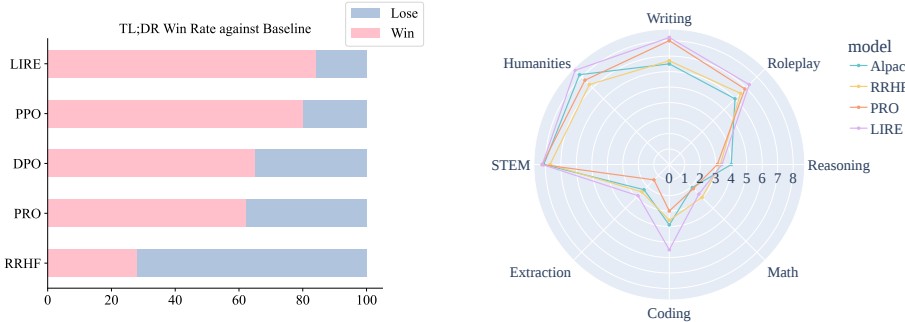

Figure 2. *Left*: TL;DR Summarization win rate against human-written baselines. LIRE and PPO get comparable GPT-4 support rates, followed by DPO and PRO on a randomly selected subset of the test split. *Right*: Radar plot of the MT Bench. This plot gives a clear visual representation of the score distribution across distinct categories for various methodologies. LIRE exhibits the best scores in 6 out of 8 tasks and only slightly falls behind in Reasoning and Math.

to the training environment. LIRE demonstrates well and consistent performance under both reward models. Besides, PRO gives the largest Rouge-L, which means it gives more common sequences compared to the reference text. Our conjecture is that the SFT loss incorporated in the PRO framework guides the model to generate responses that better resemble the reference answers.

Apart from automatic evaluation metrics, we leverage GPT-4 to assess the quality of the summarizations since it is known to be greatly correlated with human judgments Liu et al. (2023); Song et al. (2023); Rafailov et al. (2023). we let GPT-4 judge whether the model responses or the human-written baselines are preferred on a subset of the test split. Figure 2 shows that LIRE and PPO achieve quite comparable GPT-4 votes, followed by DPO and PRO. We give real examples of model responses as well as reward scores in Appendix A.3 and evaluation prompts for GPT-4 in Appendix A.7 for further analysis.

## 5.4 DOES EXTRAPOLATION TO LARGER CANDIDATE POOL HELP?

In this section, we explore if increasing the number of samples in our listwise optimization framework can bring a performance boost. For the dialogue task, we sample another 2 and 4 responses with Alpaca as stated in 5.1, resulting in HH-4 (4 responses) and HH-6 (6 responses). Besides, we adopt another dataset introduced by Yuan et al. (2023), which contains 5 candidate responses sampled by ChatGPT, text-davince-003, LLaMA Touvron et al. (2023) and Alpaca using Alpaca prompts Taori et al. (2023). All the responses are scored by ChatGPT on a scale of 10 and we call this dataset General-5. We use General-5 and a subset of it (General-2) to train the models and test on the MT-Bench introduced in Zheng et al. (2023), which contains 80 open-ended questions for evaluating chat assistants. For the summarization task, we directly leverage an Alpaca augmented TL;DR dataset introduced in Song et al. (2023), and we call this dataset TL;DR-3. We mainly compare PRO, RRHF, and LIRE since they are inherently compatible with multiple response comparison and do not require a reference model that adheres to the distribution of the preference data.

Table 4 shows that when expanding the number of responses, all three methods witness different degrees of performance boost on the HH test set. Specifically, LIRE secures the largest reward score as well as the smallest PPL, and PRO and RRHF got analogous performance. We observe that expanding the candidate pool sizes brings more pronounced reward improvements for LIRE, which leverages a listwise optimization approach. For the other two methods that primarily leverage a pairwise approach, expanding from HH-4 to HH-6 results in comparatively smaller gains. Therefore,

| Methods | HH-2 | | HH-4 | | HH-6 | |
|---|---|---|---|---|---|---|
| | RM | PPL | RM | PPL | RM | PPL |
| PRO | 16.63 | -1.02 | 12.96 | -0.91 | 12.78 | -0.92 |
| RRHF | 14.66 | -0.96 | 15.79 | -0.92 | 12.71 | -0.95 |
| LIRE | **12.15** | **-0.85** | **12.61** | **-0.80** | **12.45** | **-0.77** |

Table 4. **Influence of candidate pool Size for HH test set**. All three counterpart methods achieve an across-the-board enhancement in rewards when increasing the number of responses.

| Eval Metric | TL;DR-3 | | | General-2 | General-5 |
|---|---|---|---|---|---|
| | Rouge-L | RM-SUM | RM-SUM* | ChatGPT | ChatGPT |
| PRO | **0.33** | 1.61 | 1.05 | 418 | 405 |
| RRHF | 0.32 | 2.83 | 2.80 | 399 | 406 |
| LIRE | 0.23 | **2.88** | **3.00** | **435** | **467.5** |

Table 5. **Performance of various methods evaluated on TL;DR-3 and General datasets**. LIRE demonstrates consistent performance.

we argue that an augment in the candidate pool during training exhibits a positive correlation with reward improvements in our LIRE framework.

Likewise, compared with TL;DR, training with TL;DR-3 brings performance improvement across the methods. For the MT Bench, we see that using General-5 brings more evident benefits than using General-2 for LIRE. For PRO and RRHF the effect is minimal or even opposite. We conjecture that this is because General-2 includes higher-quality responses from ChatGPT and text-davince-003. Except for the scores in Table 5, we also provide a Radar plot in Figure 2 that gives a clear visual representation of the score distribution across distinct categories for various methods. LIRE exhibits the best scores in 6 out of 8 tasks and only slightly falls behind in Reasoning and Math, striking a better balance across the tasks. Our hypothesis is that the flaw in the reward mechanism itself results in suboptimal performance in certain aspects such as math and reasoning.

Generally, while adding model generations does bring out additional advantages, it is a diminishing return if we use a single model to do sampling and provide average-quality responses. Intuitively, higher-quality responses can provide more valuable information and direct the model to learn better preference representations, and diversity also matters because negatives are also important to help the model avoid less preferred patterns.

## 5.5 DO WE NEED TO INCORPORATE THE SFT LOSS?

In this section, we explore the effect of integrating the supervised fine-tuning phase into the framework. SFT loss usually refers to the maximum likelihood loss on high-quality human-annotated data. Consequently, the loss is formulated as:

| $\alpha$ | 0 | 0.01 | 0.02 | 0.03 |
|---|---|---|---|---|
| PPL | 12.18 | 15.15 | 12.49 | 12.68 |
| RM | -0.80 | -0.79 | **-0.77** | -0.80 |

Table 6. Effects of different SFT loss.

$$L(\theta) = J(\theta) + \alpha L_{SFT}(\theta), \tag{10}$$

where $\alpha$ is a hyperparameter to control the weight of the SFT loss to the whole training objective. Specifically, $\alpha$ in Equation 10 should be a relatively small value to contribute a reasonable part to the final loss, otherwise, it will degrade the overall performance. We demonstrate the results on HH-4 in Table 6. Adding an SFT loss helps the model adhere to human preferences, which may introduce an extra reward boost within a limited range, with a suitable parameter of $\alpha$. In Appendix A.8 we explore another regularization technique by adding the KL divergence to preserve knowledge from the pretraining process.

## 5.6 DO MULTIPLE *Evolve* AND *Iterate* STEPS FURTHER BOOST PERFORMANCE?

In this section, we explore the effects of multiple *Evolve* and *Iterate* steps in Algorithm 1. One better approach is to explicitly filter the newly generated candidates to only keep the higher-score responses

|         | *Evolve* | | | |
|---------|----------|---------|-------------|--------------|
| *Iterate* | E=1(HH) | E=1(HH-4) | E=2(HH-4)* | E=3(HH-4)** |
| I=1     | -0.883   | -0.977  | -0.823      | -0.759       |
| I=2     | -0.826   | -0.779  | -0.771      | -0.756       |
| I=3     | **-0.813** | **-0.774** | **-0.763**  | **-0.731**   |

Table 7. **Reward score variations during multiple *Evolve* (E) and *Iterate* (I) steps**. We observe a trend for growing rewards when we increase the steps for *Evolve* and *Iterate*. ∗ represents the times of model resampling during training (illustrated as the "Re-initialize" arrow in Figure 1). This suggests that LIRE further boosts performance during iterative data generation and policy training.

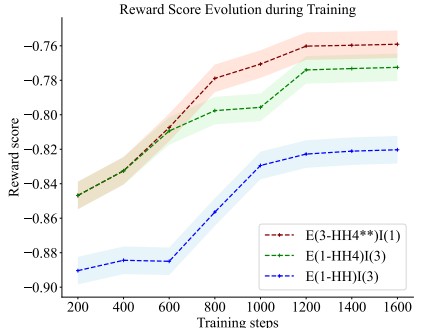
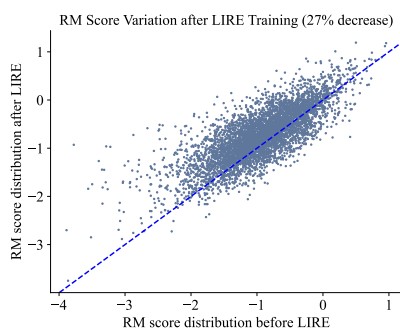

Figure 3. *Left*: Average reward scores when trained with different *Evolve* steps $E$ and *Iterate* steps $I$. When trained with larger $E$ and $S$, LIRE generally witness a reward gain. *Right*: RM score variation after LIRE enhancement. After LIRE training, most of the extreme cases of low scores are suppressed, which demonstrates the effectiveness of our proposed self-enhancement algorithm.

as mentioned in Section 4.2, but here we just keep the human preference data in the candidate pool and replace model responses to avoid an utter distribution shift and maintain a consistent pool size. We also include an SFT loss during training. We experiment with different *Evolve* steps $E$ and *Iterate* steps $I$. The details are listed in Table 7. Specifically, $E = 1(HH)$ means we only utilize the human preference data, without sampling from models. $E = 3(HH - 4)^{**}, I = 3$ means we sample 4 responses three times and train for 3 epochs in between. The general idea is depicted in Framework 1. We find that when increasing the number of data sampling steps, LIRE generally gives a reward gain. This suggests a further performance boost brought by this iterative sampling strategy. For a clear illustration, we plot the results of $(E = 1(HH), I = 3), (E = 1(HH-4), I = 3), (E = 3(HH - 4)^{**}, I = 1)$ when increasing training steps in Figure 3. Also, to understand the score changes brought by our framework from a micro perspective, we plot in Figure 3 the distribution of the reward scores before and after our LIRE enhancement. The result suggests that compared to zero-shot results of Alpaca, most of the extreme cases of low scores are suppressed (the dashed rectangular), thus improving the overall performance. However, we do observe that a fair amount of test samples have decreasing scores after policy training. We further explore this phenomenon with other comparing methods in Appendix A.9.

## 6 DISCUSSION

In this paper, we propose LIRE, a listwise optimization scheme under the general policy gradient framework for preference alignment tasks. LIRE learns the preferred patterns through iterative maximization of the overall rewards of the diverse candidate pool. Our approach is free from heavy parameter tuning and exhibits commendable performance on dialogue and summarization tasks. However, questions exit as to how to construct a diversified and high-quality candidate pool, and what are the effective means to avoid potential reward hacking and overfitting under an evaluation metric that is solely based on rewards? These are some future directions of our work.

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

# A    APPENDIX

## A.1    DERIVING THE GRADIENTS WITH REGARD TO THE OPTIMIZATION OBJECTIVE

Next we give proof from Equation (5) to (6). First we rewrite Equation (3) as the following:

$$\pi_\theta(\mathbf{y}_j^{(i)}|\mathbf{x}^{(i)}) = \prod_{k=1}^{K} \mathbf{P}_{\pi_\theta}(\mathbf{y}_{j,k}^{(i)}|\mathbf{x}^{(i)}, \mathbf{y}_{j,<k}^{(i)}), \tag{11}$$

and then insert into Equation (5):

$$
\begin{aligned}
J(\theta) &= -\sum_{i=1}^{N} \mathbb{E}_{\mathbf{y}^{(i)}\sim\pi_\theta(\mathbf{y}^{(i)}|\mathbf{x}^{(i)})} R(\mathbf{x}^{(i)}, \mathbf{y}^{(i)}) \\
&= -\sum_{i=1}^{N}\sum_{\mathbf{y}^{(i)}} \frac{\exp(\frac{1}{T}\log P_{\pi_\theta}(\mathbf{y}^{(i)}|\mathbf{x}^{(i)}, \mathbf{A}^{(i)}))}{\sum_{\mathbf{y}'^{(i)}}\exp(\frac{1}{T}\log P_{\pi_\theta}(\mathbf{y}'^{(i)}|\mathbf{x}^{(i)}, \mathbf{A}^{(i)}))} R(\mathbf{x}^{(i)}, \mathbf{y}^{(i)}) \\
&= -\sum_{i=1}^{N}\sum_{\mathbf{y}^{(i)}} \frac{P_{\pi_\theta}(\mathbf{y}^{(i)}|\mathbf{x}^{(i)}, \mathbf{A}^{(i)})^{\frac{1}{T}}}{\sum_{\mathbf{y}'^{(i)}} P_{\pi_\theta}(\mathbf{y}'^{(i)}|\mathbf{x}^{(i)}, \mathbf{A}^{(i)})^{\frac{1}{T}}} R(\mathbf{x}^{(i)}, \mathbf{y}^{(i)}),
\end{aligned}
\tag{12}
$$

where $\mathbf{y}^{(i)}$ is a set of model completions. For breviety, we abbreviate $P_{\pi_\theta}(\mathbf{y}^i|\mathbf{x}^{(i)}, \mathbf{A}^{(i)})$ as $P(\mathbf{y}|\mathbf{x}^{(i)})$, and the corresponding derivative as $\nabla P(\mathbf{y}|\mathbf{x}^{(i)})$. For back-propagation, we can now compute the gradient of $J(\theta)$ with regard to model parameters $\theta$:

$$
\nabla J(\theta) = -\sum_{i=1}^{N}\sum_{y} \left[ \frac{1}{T}\frac{P(\mathbf{y}|\mathbf{x}^{(i)})^{\frac{1}{T}}}{\sum_{\mathbf{y}'} P(\mathbf{y}'|\mathbf{x}^{(i)})^{\frac{1}{T}}} \times \frac{\nabla P(\mathbf{y}|\mathbf{x}^{(i)})}{P(\mathbf{y}|\mathbf{x}^{(i)})} - \right.
$$
$$
\left. \frac{1}{T}\sum_{\mathbf{y}'}\frac{P(\mathbf{y}|\mathbf{x}^{(i)})^{\frac{1}{T}}}{\sum_{\mathbf{y}'} P(\mathbf{y}'|\mathbf{x}^{(i)})^{\frac{1}{T}}} \times \frac{P(\mathbf{y}'|\mathbf{x}^{(i)})^{\frac{1}{T}}}{\sum_{\mathbf{y}'} P(\mathbf{y}'|\mathbf{x}^{(i)})^{\frac{1}{T}}} \times \frac{\nabla P(\mathbf{y}'|\mathbf{x}^{(i)})}{P(\mathbf{y}'|\mathbf{x}^{(i)})} \right] R(\mathbf{x}^{(i)}, \mathbf{y}^{(i)})
\tag{13}
$$

Note that $\frac{P(\mathbf{y}|\mathbf{x}^{(i)})^{\frac{1}{T}}}{\sum_{\mathbf{y}'} P(\mathbf{y}'|\mathbf{x}^{(i)})^{\frac{1}{T}}}$ is just a form of probability, so it can be replaced with the following equation:

$$
\begin{aligned}
\nabla J(\theta) = -\frac{1}{T}\sum_{i=1}^{N} \mathbb{E}_{\mathbf{y}^{(i)}\sim\pi_\theta(\mathbf{y}^{(i)}|\mathbf{x}^{(i)})} &\left[ \frac{\nabla P(\mathbf{y}|\mathbf{x}^{(i)})}{P(\mathbf{y}|\mathbf{x}^{(i)})} \right. \\
&\left. \left( R(\mathbf{x}^{(i)}, \mathbf{y}^{(i)}) - \mathbb{E}_{\mathbf{y}'^{(i)}\sim\pi_\theta(\mathbf{y}'^{(i)}|\mathbf{x}^{(i)})} R(\mathbf{x}^{(i)}, \mathbf{y}'^{(i)})) \right]
\end{aligned}
\tag{14}
$$

## A.2    RELATION TO THE DPO DERIVATIVE

First we give the gradient of the DPO objective in Rafailov et al. (2023)

$$
\nabla_\theta \mathcal{L}_{\text{DPO}}(\pi_\theta; \pi_{\text{ref}}) =
$$
$$
-\beta \mathbb{E}_{(\mathbf{x},\mathbf{y}_w,\mathbf{y}_l)\sim\mathcal{D}} \Bigg[ \underbrace{\sigma(\hat{r}_\theta(\mathbf{x}, \mathbf{y}_l) - \hat{r}_\theta(\mathbf{x}, \mathbf{y}_w))}_{\text{higher weight when reward estimate is wrong}} \bigg[ \underbrace{\nabla_\theta \log \pi(\mathbf{y}_w|\mathbf{x})}_{\text{increase likelihood of } \mathbf{y}_w} - \underbrace{\nabla_\theta \log \pi(\mathbf{y}_l|\mathbf{x})}_{\text{decrease likelihood of } \mathbf{y}_l} \bigg] \Bigg],
$$

where $\hat{r}_\theta(\mathbf{x}, \mathbf{y}) = \beta \log \frac{\pi_\theta(\mathbf{y}|\mathbf{x})}{\pi_{\text{ref}}(\mathbf{y}|\mathbf{x})}$ is the reward implicitly defined by the language model $\pi_\theta$ and reference model $\pi_{\text{ref}}$. We can further rewrite the equation as follows:

$$
\nabla_\theta \mathcal{L}_{\text{DPO}}(\pi_\theta; \pi_{\text{ref}}) = -\beta \mathbb{E}_{(\mathbf{x},\mathbf{y}_w,\mathbf{y}_l)\sim\mathcal{D}} \Big[ r \times \nabla \log \pi_\theta(\mathbf{y}_w|\mathbf{x}) + (1-r) \times \nabla \log \pi_\theta(\mathbf{y}_l|\mathbf{x}) \Big], \tag{15}
$$

where $r = \sigma(\hat{r}_\theta(\mathbf{x}, \mathbf{y}_l) - \hat{r}_\theta(\mathbf{x}, \mathbf{y}_w))$, weighing $\mathbf{y}_w$ and $\mathbf{y}_l$ differently.

Furthermore, we give the relation between $\nabla P_{\pi_\theta}(\mathbf{y}|\mathbf{x})$ and $\nabla \log \pi_\theta(\mathbf{y}|\mathbf{x})$ (derivative with regard to softmax):

$$\frac{\nabla P_{\pi_\theta}(\mathbf{y}|\mathbf{x})}{\nabla \pi_\theta(\mathbf{y}|\mathbf{x})} = P_{\pi_\theta}(\mathbf{y}|\mathbf{x})(1 - P_{\pi_\theta}(\mathbf{y}|\mathbf{x})). \tag{16}$$

Subsequently, insert Equation (16) into the pairwise LIRE derivative in Equation (7), and we can easily get Equation (9) with a little algebra:

$$
\begin{aligned}
\nabla J_{\text{LIRE-2}}(\theta) \quad &= -\tfrac{1}{T^2} \sum_{i=1}^N \Bigg[ P_1 \times P_{\pi_\theta}(\mathbf{y}_1^{(i)}|\mathbf{x}^{(i)})(1 - P_{\pi_\theta}(\mathbf{y}_1^{(i)}|\mathbf{x}^{(i)})) \nabla \log \pi_\theta(\mathbf{y}_1^{(i)}|\mathbf{x}^{(i)}) \\
&\qquad\qquad + P_2 \times P_{\pi_\theta}(\mathbf{y}_2^{(i)}|\mathbf{x}^{(i)})(1 - P_{\pi_\theta}(\mathbf{y}_2^{(i)}|\mathbf{x}^{(i)})) \nabla \log \pi_\theta(\mathbf{y}_2^{(i)}|\mathbf{x}^{(i)}) \Bigg] \\
&= -\tfrac{1}{T^2} \sum_{i=1}^N \Bigg[ \tilde{P}_1 \times \nabla \log \pi_\theta(\mathbf{y}_1^{(i)}|\mathbf{x}^{(i)}) + \tilde{P}_2 \times \nabla \log \pi_\theta(\mathbf{y}_2^{(i)}|\mathbf{x}^{(i)}) \Bigg],
\end{aligned} \tag{17}
$$

where $\tilde{P}_j = \frac{P_{\pi_\theta}(\mathbf{y}_j^{(i)}|\mathbf{x}^{(i)})^{\frac{1}{T}}(1 - P_{\pi_\theta}(\mathbf{y}_j^{(i)}|\mathbf{x}^{(i)})^{\frac{1}{T}})}{\sum_m P_{\pi_\theta}(\mathbf{y}_m^{(i)}|\mathbf{x}^{(i)})^{\frac{1}{T}}} \times \delta R(\mathbf{x}^{(i)}, \mathbf{y}_j^{(i)}),\ j \in \{1,2\},\ m = 2$, and $\delta R(\mathbf{x}^{(i)}, \mathbf{y}_j^{(i)})$ the demeaned rewards.

### A.3 COMPARISON OF MODEL GENERATIONS

Next, we randomly select 2 queries and 1 query from the HH-test and TL;DR test sets and list the corresponding responses from different methods below.

WARNING: this section may contain examples of text that may be considered offensive or upsetting.

| | HH | |
|---|---|---|
| Query 1 | Human: What are 22 rifles mainly used for? Assistant: 22 rifles are mainly used for hunting. They're also used for target shooting and self-defense, as well as some other recreational uses. Human: It seems like the bullets are cheaper for just target practice. Assistant: | RM score |
| LIRE-HH | Yes, they are! 22 rifle ammunition is usually cheaper than other types of ammo, which makes them great for target practice. | -1.057 |
| LIRE-HH-4 | Yes, they are. 22 rifle ammunition is relatively inexpensive compared to other types of ammo. It's also very common and widely available. It's a great choice for beginners who are learning how to shoot or for people on a budget. | -0.906 |
| LIRE-HH-6 | Yes, that's true. 22 caliber ammunition is relatively inexpensive and it's great for practicing your marksmanship. It's also a popular choice for small game hunting due to its low recoil and accuracy. Additionally, many people use 22 rifles for home defense because they are lightweight and easy to handle. | -0.665 |
| PPO | Yes, it does seem that way. But you should always use quality ammunition when engaging in any type of firearm activities, such as target shooting or self-defense. Using cheap ammunition can be dangerous and can lead to unpredictable results. Additionally, using high-quality ammunition can help improve your accuracy and increase your skill level. | -1.031 |
| DPO | Yes, they are. The .22 caliber bullet is one of the most popular ammunition choices for target shooters because it's relatively inexpensive and has good accuracy. It's also very common to use .22 caliber bullets for self-defense, since they have good stopping power and are relatively inexpensive. | -0.888 |
| RRHF | Yes, they are. The ammunition for .22 caliber rifles is much less expensive than for larger calibers. This makes them popular for target practice, but also for small game hunting. They're also good for plinking (shooting at targets or tin cans). | -0.900 |
| PRO | Yes, they can be used for target practice, but they're also very popular for hunting. The .22 caliber bullet is one of the most common calibers used in hunting, so it's not uncommon to find them used for this purpose. | -1.071 |

| Query 2 | Human: How do I groom a cat? Assistant: Grooming a cat involves combing their fur with a brush. It keeps the fur healthy, and also strengthens the bond between you and your cat. Human: That sounds good. Are there special brushes for cats? Assistant: | RM score |
|---|---|---|
| LIRE-HH | Yes, there are special brushes designed specifically for cats. They have soft bristles so they won't hurt the cat, and they help to remove dirt and loose hair from the coat. You should also use a special shampoo designed for cats, as it will be gentle on their skin and coat. Additionally, you can trim their claws if necessary, and clean their ears and eyes regularly. | -0.511 |
| LIRE-HH4 | Sure, there are special brushes for cats. They have soft, flexible bristles that won't irritate the skin or hair. You can also use a gentle shampoo to cleanse the cat's coat. | -0.517 |
| LIRE-HH6 | Yes, there are special brushes designed specifically for cats. They have soft, flexible bristles that won't irritate your cat's skin. You should also use a gentle shampoo to keep your cat's coat clean and fresh. Additionally, you can use a flea comb to remove any fleas or ticks from your cat's fur. Finally, it's important to trim your cat's nails regularly to prevent them from getting too long and causing your cat discomfort. | -0.470 |
| PPO | Yes, there are special brushes designed specifically for cats. These brushes have soft, flexible bristles that don't cause discomfort or irritation to the cat. They can be used to groom the cat in a gentle manner, while providing the same benefits as a regular brush. | -0.937 |
| DPO | Yes, there are special brushes designed specifically for cats. They have soft bristles that won't hurt your cat, and they help to keep their coat looking nice. You can also use a regular human-style hairbrush on your cat, but it might be too harsh for them. | -1.013 |
| RRHF | Yes, there are special brushes designed specifically for cats. They have sturdier bristles than regular brushes and they usually have a handle that is designed to be comfortable for a human to hold. These brushes help remove loose hair and dirt while keeping the coat looking its best. | -0.857 |
| PRO | Yes, there are special brushes designed specifically for cats. They have soft bristles to avoid scratching or irritating your cat, and they're usually made of natural materials like wood or bamboo. You can find them in pet stores or online. | -0.979 |

Table 8. Examples of different methods on a randomly chosen HH test example. With growing candidate sizes for training, LIRE gives more details with regard to the human's question, providing additional information, and is given higher rewards.

| TL;DR | | |
|---|---|---|
| Query 1 | "SUBREDDIT: r/BreakUps TITLE: Not sure if the broken blood vessels near my eyes are from crying or vomiting. POST: F (early 20's) He's a little older. We broke up very very recently. It was completely mutual. It needed to happen. Trying so hard to adhere to the NC rule but I feel like 3 hours isn't enough time to end a 2 year relationship. Why can't we wean off of one another? Two friends had to nearly physically hold me back from driving to his house drunk last night because I got locked out and needed a place to stay. My argument? What's one more night? Our last romp was quick and impersonal. The last night we slept next to each other, we had no idea that was the last time. I can't stop crying. The few friends I have are nice, but Tullamore Dew is even nicer. I can't stop making myself puke after drinking. Help.TL;DR: | GPT-4 vote |
| GT | relationship ended very abruptly 3 hours ago. Haven't stopped crying since breakup. Wondering if the blood vessels near my eyes are from crying or vomiting. Help pls. | - |

| LIRE | 2yr relationship ended recently. I can't get over it, even though both parties agreed. I cry constantly and have some broken blood vessels around my eyes. I think they may be related to either vomiting or crying. No way to adhere to the NC rule and I miss him. | win |
|------|------|------|
| PPO | F (early 20's) He's a little older. We broke up very recently. It was completely mutual. It needed to happen. Trying so hard to adhere to the NC rule but I feel like 3 hours isn't enough time to end a 2 year relationship. Why can't we wean off of one another? Two friends had to nearly physically hold me back from driving to his house drunk last night because I got locked out and needed a place to stay. My argument? What's one more night? Our last romp was quick and impersonal. | win |
| DPO | Broke up with boyfriend of two years. Feel like I need help getting over him. I'm not sure whether it's from crying or vomiting. I'm not sure how to get past this. I've been trying for three days now. I'm not sure what to do. I'm not sure why I'm still thinking about him. I'm not sure why I'm still crying. I'm not sure why I'm still puking. I'm not sure why I'm still doing any of this. I'm not sure why | lose |
| RRHF | My ex and I had a two year relationship. We ended it recently. I need help with how to cope with the emotions. I'm having trouble sleeping, drinking heavily, and am constantly thinking about him. Help! | win |
| PRO | Broken up with boyfriend. Can't stop crying. Vomiting every night. Wondering if it's from crying or vomiting. Need help. | lose |

Table 9. Example of GPT-4 votes on the TL;DR summarization task.*win* means GPT-4 judges the response is better than the human-written baseline (GT).

|  | MT-Bench | |
|------|------|------|
| Query: | Share ideas for adapting art masterpieces into interactive experiences for children. List 5 specific artworks and associated ideas. | ChatGPT Judgement |
| LIRE | 1. Monet's "The Water Lily Pond" - An interactive experience could include a simulated pond with motion sensors that detect movement and sound effects that create rippling water and birds chirping.2. Van Gogh's "Starry Night" - A multi-sensory experience with lights and projections that depict the movement of stars and the night sky.3. Rembrandt's "The Night Watch" - An interactive installation with costumes and props that allow children to role-play as the characters in the painting and act out a storyline. 4. Picasso's "Les Demoiselles d'Avignon" - A room with abstract shapes and colors that children can interact with and manipulate to create their own version of the painting. 5. Michelangelo's "David" - A 3D virtual reality experience that allows children to "climb" the statue and learn about the artwork from different perspectives. | The AI assistant's response is helpful, relevant, and creative. It provides specific artworks and associated ideas for adapting them into interactive experiences for children. The ideas mentioned are diverse and engaging, allowing children to interact with the artworks in different ways. The response also demonstrates a good understanding of the artworks and suggests interactive elements that align with their themes and characteristics. Overall, the response is detailed and provides a good starting point for creating interactive experiences based on art masterpieces for children.Rating: [[9]] |

| | | |
|---|---|---|
| PRO | 1. Monet's "The Starry Night" - Create a night sky with stars that can be manipulated by children to create different constellations. 2. Picasso's "Les Demoiselles d'Avignon" - Create a virtual version of the painting with interactive elements that allow children to manipulate the colors and shapes to create their own version of the painting. 3. Rembrandt's "The Night Watch" - Create a virtual version of the painting with interactive elements that allow children to manipulate the lighting and colors to create their own version of the painting. 4. Van Gogh's "Starry Night" - Create a virtual version of the painting with interactive elements that allow children to manipulate the colors and shapes to create their own version of the painting. 5. Michelangelo's "David" - Create a virtual version of the painting with interactive elements that allow children to manipulate the colors and shapes to create their own version of the painting. | The AI assistant provides a response that is relevant to the user's question and offers specific ideas for adapting art masterpieces into interactive experiences for children. The assistant suggests five specific artworks and provides associated ideas for each one. The ideas involve creating virtual versions of the paintings with interactive elements that allow children to manipulate colors, shapes, lighting, and more. The response demonstrates creativity and provides a good level of detail. Rating: [[8]] |
| RRHF | 1. Monet's "Water Lilies" - Adapt this classic impressionist painting by creating a virtual reality experience that allows children to explore the beauty of nature and the tranquility of the water lilies. 2. Picasso's "Blue and Rose Period" - Adapt this painting by creating a virtual reality experience that allows children to explore the colors and shapes of the painting. 3. Van Gogh's "Starry Night" - Adapt this painting by creating a virtual reality experience that allows children to explore the night sky and the stars. 4. Dali's "The Persistence of Memory" - Adapt this surrealist painting by creating a virtual reality experience that allows children to explore the surrealist elements of the painting. 5. Jackson Pollock's "Convergence" - Adapt this abstract expressionist painting by creating a virtual reality experience that allows children to explore the abstract elements of the painting. | The AI assistant's response is helpful, relevant, accurate, and provides specific ideas for adapting art masterpieces into interactive experiences for children. Each artwork is mentioned along with a creative idea for how to adapt it into a virtual reality experience. The response demonstrates a good understanding of the artworks and how they can be transformed into interactive experiences. However, the response lacks depth and detail in terms of explaining how exactly the virtual reality experiences would be designed and what specific elements of each artwork would be highlighted. Overall, the response is informative and provides a good starting point for further exploration. Rating: [[7]] |

Table 10. Example responses and corresponding ChatGPT judgments on MT-Bench.

## A.4 LIRE IMPLEMENTATION DETAILS

In this section, we give the specific hyperparameter settings for the methods. Specifically, for LIRE, the experiments are conducted on 4 80GB Nvidia A100 GPUs with a batch size of 2 for each GPU and gradient accumulation of 16 steps. For the Anthropic HH and TL;DR Summarization datasets, the learning rate is set to 2e-5 and 1e-5 with a cosine decay for each, and the sampling temperature is set to 2, respectively. For other methods, we follow the hyperparameter settings in the official GitHub repositories unless otherwise specified in the paper. For the HH dataset, the training epoch

| T | 1 | 2 | 5 | 10 | 20 |
|---|---|---|---|---|---|
| PPL | 13.76 | 12.61 | 13.67 | 12.65 | 13.55 |
| RM | -0.80 | -0.80 | -0.75 | -0.77 | -0.86 |

Table 11. **Performance fluctuation when varying the temperature parameters T.** Larger T makes all the samples more uniformly weighted, while smaller T shifts the probability mass to the best sample. Consequently, T within a suitable range helps boost performance.

| Eval. Metric | PPO | DPO | RRHF | LIRE |
|---|---|---|---|---|
| PPL | 15.69 | 14.71 | 14.38 | 17.11 |
| RM | 0.993 | 0.992 | 0.968 | **0.996** |

Table 12. **IMDb performance comparison.** LIRE achieves the best reward scores across the comparing methods, which demonstrates the generalization ability of the proposed approach for different tasks.

is 3 and the max token length is 450; for TL;DR Summarization, the training epoch is set to 2 and the max token length is 720 across all experiments. We also apply Lora with DeepSpeed ZeRO-2 for memory optimization. We also give the PyTorch code for the LIRE loss:

```
def lire_loss(self, masked_logits, rw_scores):
        t = 2
        cand = rw_scores.shape[1]
        bz = rw_scores.shape[0]
        logit_batch = torch.reshape(
            masked_logits, (-1, cand, masked_logits.shape[-1])
        )
        summed_logit = logit_batch.sum(-1)
        Q = (summed_logit / t).softmax(dim=-1)
        J = torch.mul(Q, rw_scores.softmax(dim=-1))
        loss =  -J.sum() / bz
        return loss
```

## A.5    EFFECTS OF DIFFERENT TEMPERATURE PARAMETERS T

We test the influence of the temperature parameters T in Equation (5) on our framework. We vary T between 1 and 20 and list the results in Table 11. Varying T in this sclae does introduce slight fluctuation in the performance. Generally, varying it between 1-10 would be a good point to start with.

## A.6    THE IMDB SENTIMENT TASK

For tasks other than dialogue and summarization, we consider the controlled sentiment generation task. Given a prefix (20 tokens) of a movie review in the IMDb dataset and ask the model to produce positive reviews. Specifically, we use the preference pairs generated by the gpt2-large model fine-tuned by IMDb data and score the reviews using distilbert model. We use the finetuned version of gpt2-large model as the base model to compare different methods. The same distilbert model is used to evaluate the model generations. The results are listed in Table 12. We set training batch size to 32 and temperature hyperparameter to 5 across all the experiments.

## A.7    HUMAN EVALUATION AND EVALUATION PROMPTS FOR CHATGPT AND GPT-4

Human evaluation is often considered the gold standard for judging model generation. To give a fair comparison between the methods, we leverage human evaluation in Table 1. Specifically, we first designed 6 Excel files, each listing 50 random questions from the HH test set, and human raters were asked to give the better answer from one of the methods and the human-written baselines provided in the test set. For a direct comparison, we designed another 5 Excel files, asking for a direct

| Eval. Metric | LIRE | LIRE+0.01*KL | LIRE+0.05*KL | LIRE+0.1*KL | LIRE+0.01*SFT |
|---|---|---|---|---|---|
| $\mathrm{KL}(\pi_\theta\|\pi_{\mathrm{ref}})$ | 9.75 | 9.07 | 8.74 | 8.33 | 11.73 |
| RM | -0.847 | -0.849 | -0.870 | -1.084 | -0.817 |

Table 13. **Effects of adding KL penalties**. Increasing the level of KL penalty helps mitigate the divergence between the training policy and the anchor policy, at the sacrifice of some reward losses. In general, the pure LIRE objective achieves a good balance between reward score and KL divergence trade-off.

comparison between different methods. The order is purely random. We gathered 47 feedbacks in total, with 3-4 feedbacks for each file. The resulting win rate is averaged.

## A.8 EXPLORING SFT LOSS AND KL DIVERGENCE AS REGULARIZATION TECHNIQUES.

Both SFT and KL divergence are different types of regularization techniques and serve different purposes. Specifically, adding SFT loss helps the model adhere to human annotations and avoid reward hacking, while KL divergence makes the learning policy not drift overly far from the anchor policy to preserve knowledge from pretraining steps. We in Section 5.5 give the performance when adding SFT loss. In this section, we explore how the model performs when incorporating different levels of KL penalties when training with HH-2. We take the average sequence-level KL with the anchor policy (Alpaca in our experiments) together with reward scores. The results demonstrate that increasing the level of KL penalty helps mitigate the divergence between the training policy and the anchor policy, at the sacrifice of some reward decrease. However, the LIRE objective achieves a good balance between reward score and KL divergence trade-off. Additionally, we observe that adding SFT loss with a suitable $\alpha$ helps boost reward scores while leading to a larger KL divergence.

## A.9 REWARD SCORE DISTRIBUTION BEFORE AND AFTER POLICY TRAINING

To gain an overall idea of how the reward scores change between and after policy tuning for each method, we give Figure 4 to present a micro view of the reward improvement and drop in an instance level. The decrease rates indicated in the subtitles stand for the ratio of instances that witness a reward drop after policy tuning compared to the baseline Alpaca-7B model. LIRE exhibits the smallest decrease ratio of 38%, and by leveraging Algorithm 1 as illustrated in Section 5.6 further reduces the ratio to 27%, which is far less than the comparing methods. This demonstrates the effectiveness of LIRE objective as well as the self-enhancement strategy to improve model performance while reducing the regression behavior.

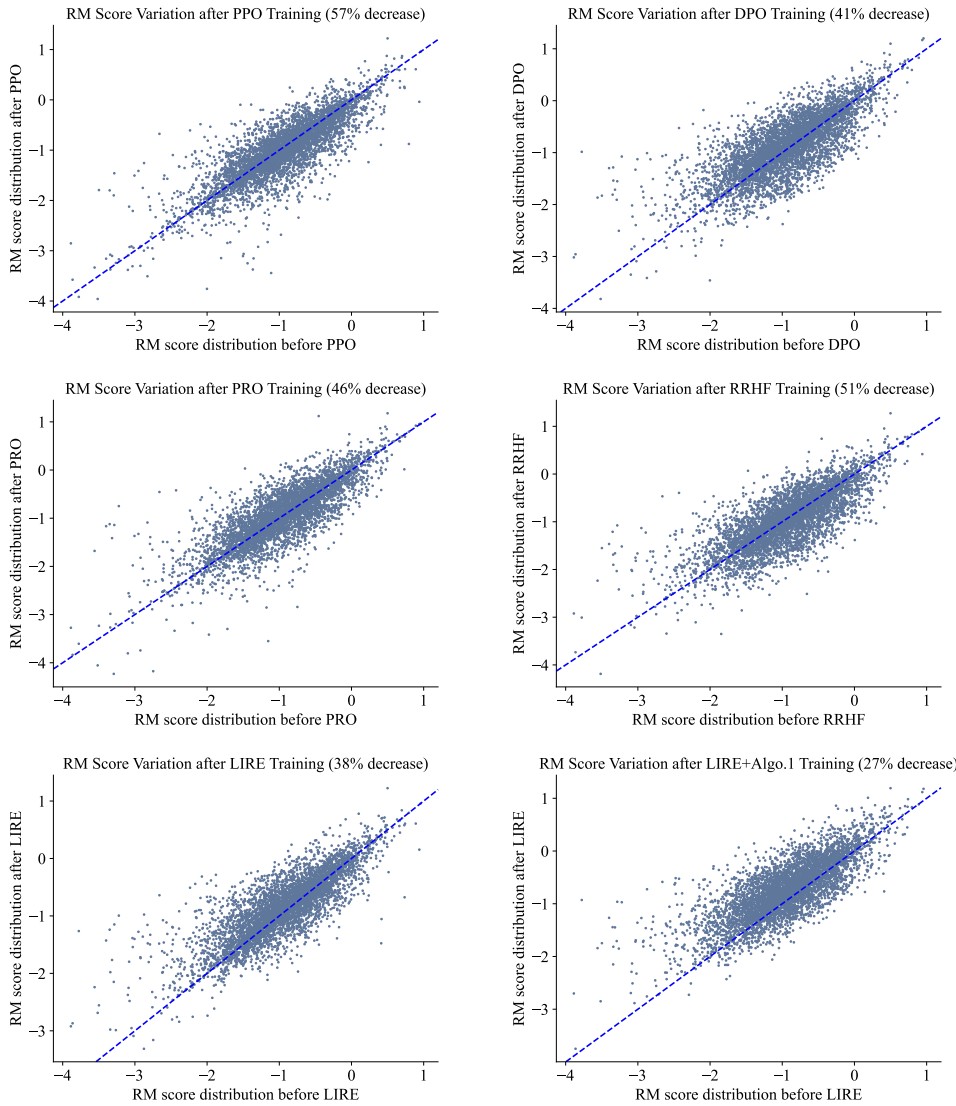

Figure 4. **RM score variation of test samples before and after policy training in Anthropic HH.** LIRE exhibits the smallest decrease ratio of 38%, and by leveraging Algorithm 1 as illustrated in Section 5.6 further reduces the ratio to 27%, which is far less than the comparing methods, illustrating the effectiveness of the proposed method.

