# OpenReview forum: "LIRE: Listwise Reward Enhancement for Preference Alignment"
_ICLR.cc/2024/Conference — Submitted to ICLR 2024_

### Official Review · Reviewer_XkkK · 2023-11-01

**Soundness:** 2 fair
**Presentation:** 3 good
**Contribution:** 2 fair
**Rating:** 5
**Confidence:** 3

**Summary:**

In this paper, the authors propose a new training method for language model reinforcement finetuning. Instead of conducting pairwise comparisons as in the conventional RLHF setting, they propose to sample multiple responses and score each of them with their log-likelihood. After that, they assign the probability for each sample response with the score and maximize the expected reward given by the learned reward model. They found that the method has a close connection with direct preference optimization (DPO). They conduct experiments on HH and TL;DR datasets to confirm the effectiveness of their methods.

It is interesting to see a new approach proposed for RLHF with a tight connection with previous methods. However, the paper may have the following flaws:
1. The LIRE objective is essentially the original policy gradient objective. In Equation (5), the author have $$J(\theta) = -\sum_{x}\sum_{y} P_{\pi_\theta}(y|x, A) R(x, y),$$ where $P_{\pi_\theta} (y|x, A) \propto \exp( \frac{1}{T} \sum_k \log P(y_k | x))$. The problem arises when $T=1$, we then have $P_{\pi_\theta} (y|x, A) = \prod_k P(y_k | x) = P(y|x)$. This suggests that $$J(\theta) = -\sum_{x}\sum_{y} P_{\pi_\theta}(y|x) R(x, y),$$ which is equivalent sampling multiple trajectories in RL for each query. It looks to me that the only difference from the original policy gradient objective that could be made here is choosing $T \not=1$. However, it is not well justified to have such a practice. It is also documented in Table 10 that varying $T$ introduces slight fluctuation in performance, which does not seem to have a consistent pattern.
2. The performance improvement is marginal. In Table 3 and Table 4, the performance of LIRE is only slightly better than PPO, especially when there are multiple responses sampled.

**Strengths:**

It is interesting to see a new approach proposed for RLHF with a tight connection with previous methods.

**Weaknesses:**

1. The LIRE objective is essentially the original policy gradient objective. In Equation (5), the author have $$J(\theta) = -\sum_{x}\sum_{y} P_{\pi_\theta}(y|x, A) R(x, y),$$ where $P_{\pi_\theta} (y|x, A) \propto \exp( \frac{1}{T} \sum_k \log P(y_k | x))$. The problem arises when $T=1$, we then have $P_{\pi_\theta} (y|x, A) = \prod_k P(y_k | x) = P(y|x)$. This suggests that $$J(\theta) = -\sum_{x}\sum_{y} P_{\pi_\theta}(y|x) R(x, y),$$ which is equivalent sampling multiple trajectories in RL for each query. It looks to me that the only difference from the original policy gradient objective that could be made here is choosing $T \not=1$. However, it is not well justified to have such a practice. It is also documented in Table 10 that varying $T$ introduces slight fluctuation in performance, which does not seem to have a consistent pattern.
2. The performance improvement is marginal. In Table 3 and Table 4, the performance of LIRE is only slightly better than baselines, especially when there are multiple responses sampled.

**Questions:**

Could you provide a possible explanation of why PRO and RRHF perform significantly worse when the number of candidates is small (e.g. Table 3)?

---

> ### Author Response · Authors · 2023-11-16
>
> We thank the reviewer for the detailed and constructive comments. Below are our responses to all the questions and comments, and we hope that the responses are reasonable and satisfactory enough to address the reviewer’s concerns and we welcome further discussion.
>
> **Q$_1$: "The LIRE objective is essentially the original policy gradient objective..."**
>
> $A_1$: As illustrated in Section 3, our objective is derived within the general policy gradient framework, and our focus is on how to better formulate the probability distribution over the multiple model generations during training.  We construct P in Equation 4 by taking a softmax over all model generations.  This operation makes the $P_{\pi(\theta)}(y|x, A)$ $\textbf{not}$ proportional to $\exp(\frac{1}{T})\sum_{k}logP(y_k |x)$. Consequently, when T=1, the LIRE objective still incorporates information from other responses(the denominator), which corresponds to the concept of “listwise”, and this makes it different from the policy gradient framework as the reviewer illustrated. This practice also brings a demeaned reward score
> $R(x^{(i)}, y^{(i)})-E_{(y′^{(i)}∼\pi_\theta(y^{(i)}|x^{(i)}))}R(x^{(i)}, y′^{(i)})$ in Equation 6, which brings advantages that responses with higher rewards contribute a larger part to the gradient descent.
>
> **Q$_2$: "varying T introduces slight fluctuation in performance..."**
>
> A$_2$: Essentially, the temperature parameter T is introduced to modify the probability distribution of the sampled model completions for a given query. Larger T makes all the samples more uniformly weighted, while smaller T shifts the probability mass to the best sample (more like SFT training), therefore, varying T introduces slight fluctuation in performance and T should be within a suitable range to boost performance.
>
> **$Q_3$: "The performance improvement is marginal..."**
>
> A$_3$: In original Table 3 and 4, the (number) in green indicates performance improvement of the model trained with the current training set (the column name) compared to the model trained with a dataset of fewer responses ( the previous column).  So as the size of candidate responses increases,  the performance continues to improve.  We are sorry for the confusing illustration and will remove the (number) for clarity. The performance improvement (RM score) for summarization and dialogue tasks are summarized in the following. In Table 7, we also show that the performance boost is more evident if we utilize Algorithm 1 (iteratively updating the training samples) to increase the quality of the training samples. Generally, the pattern is a growing RM score if we increase the candidate numbers, while other pairwise methods do not display this pattern.
>
> | Eval. Metric (Summarization) | baseline | LIRE trained with TL;DR | trained with TL;DR-3 |
> | ---------------------------- | -------- | ----------------------- | ----------------------- |
> | RM-SUM                       | -1.74    | 2.76                    | $\textbf{2.88}$                    |
> | RM-SUM*                      | -0.31    | 2.79                    | $\textbf{3.00}$                  |
>
> | Eval. Metric(Dialogue) | baseline | LIRE trained with HH-2 | trained with HH-4 | trained with HH-6 | Using Algo. 1 |
> | ---------------------- | -------- | ---------------------- | ----------------- | ----------------- | ------------- |
> | RM                     | -0.93    | -0.85                  | -0.80             | -0.77             | $\textbf{-0.73}$         |
>
>
> **$Q_4$: "Could you provide a possible explanation of why PRO and RRHF perform significantly worse..."**
>
> A$_4$: This is an interesting topic. The RRHF objective tries to rank the probabilities of model generations according to the rank given by reward scores, the model relies on long sequence length to incorporate more preference information; the PRO objective leverages (n-1) comparisons in a list of n responses, and for each comparison, the best example is contrasted against the "negative" examples. According to the Minimum Bayes Risk(MBR) framework [1], choosing the outputs of a machine learning system based not on the output with the highest probability, but the output with the lowest risk (largest reward) among multiple candidates can be more powerful. Both RRHF and PRO do not directly optimize for reward scores, while constructing the loss with sophisticated modeling of probability distribution in the sampled space, and thus may, in our opinion, underperform when the number of candidates is small.
>
> [1]. Bertsch, Amanda, et al. "It's MBR All the Way Down: Modern Generation Techniques Through the Lens of Minimum Bayes Risk." arXiv preprint arXiv:2310.01387 (2023).

---

> > ### Comment · Reviewer_XkkK · 2023-11-22
> >
> > Thank you for your response. For the first two questions, I am still not convinced by your answers. With $T=1$, the $\exp$ and $\log$ would cancel out and the equation essentially becomes the "batched policy gradient", which is the default practice. Consequently, the second question is important to ask, but I believe the current response does not provide clear evidence of how $T\not=1$ helps. Therefore, the rating remains unchanged.

---

> ### Author Response · Authors · 2023-11-23
>
> Thanks for your reply. The policy gradient under $m$ trajectories takes the form:
>
> $J_\theta=\frac{1}{m}\sum_{j=1}^{m} \log\pi_\theta(a_j|s_j)R(\tau_j)$
>
> so the log-probability of resposnes $\log\pi_\theta$ is multipled by $R$  to achieve optimization. On the other hand, our objective takes the form:
>
> $J_\theta=\sum_{j=1}^{m}P_{\pi_\theta}(a_j|s_j)R(\tau_j)$
>
> where $P_{\pi_\theta}$ represents the soft-max normalized log probabilities of responses as in Equation 4, which is essentially still a probability distribution itself. Even when T=1, we cancel out log and exp, resulting in $P_{\pi_\theta}=\frac{\pi_\theta(a_i|s_i)}{\sum_{j=1}^{m}\pi_\theta(a_j|s_j)}$
>
> and this is **not** equalivent to  $\frac{1}{m}\log\pi_\theta(a_j|s_j)$.
>
> In softmax policy gradient [2][3], the policy is first parametrized using $\pi_\theta(.|s)=softmax(\theta(s,.))$,  where  $\theta(s,a)$ is referred as logits for an action (a single token in our settings). However, this practice is implemented in the token space for constructing the distribution for a single sentence, and during the policy optimization, actions(tokens) are actively sampled according to this constructed sampling distribution,  with no softmax applied to multiple sentences to formulate a softmax-normalized probability distribution. Differently, in LIRE, given several candidate responses, we take a softmax over the sampled model completions(sentences) to construct a probability distribution $P$ for policy optimization.
>
> According to the results in Table 11, T=1 actually produces good results, and we do not specifically choose T $\neq$1 since this is actually a good practice. We also implemented the policy gradient equation above, leading to worse results of -1.12(RM) and 18.89 (PPL).
>
> | T    | 1     | 2     | 5     | 10    | 20    |
> | ---- | ----- | ----- | ----- | ----- | ----- |
> | PPL  | 13.76 | 12.61 | 13.67 | 12.65 | 13.55 |
> | RM   | -0.80 | -0.80 | -0.75 | -0.77 | -0.86 |
>
> [2]. Mei, Jincheng, et al. "On the global convergence rates of softmax policy gradient methods." *International Conference on Machine Learning*. PMLR, 2020.
>
> [3]. Li, Gen, et al. "Softmax policy gradient methods can take exponential time to converge." *Conference on Learning Theory*. PMLR, 2021.

---

### Official Review · Reviewer_Vfn5 · 2023-11-01

**Soundness:** 3 good
**Presentation:** 3 good
**Contribution:** 2 fair
**Rating:** 6
**Confidence:** 3

**Summary:**

The paper proposes LIRE, a listwise optimization framework for aligning language model generations with human preferences. The key contributions are:

1. Formulates preference alignment as directly optimizing rewards in a listwise manner over multiple candidate responses. This avoids explicit pairwise comparisons.

2. Proposes a listwise softmax loss that incorporates reward scores into the training objective. Higher rewards are encouraged while lower ones are depressed.

3. Introduces a self-enhancement algorithm with iterative data sampling and policy updates to progressively improve reward.

4. Experiments show LIRE achieves superior and consistent performance on dialogue and summarization tasks. Benefits increase as candidate pool size grows.

5. Analysis provides insights into LIRE's derivatives and relation to other preference learning methods like DPO. Overall, it demonstrates an effective listwise approach to preference alignment.

**Strengths:**

Here are some key strengths of the LIRE paper:

1. Flexible training framework: LIRE loss neatly incorporates rewards into the objective. Self-enhancement via iterative sampling boosts performance. Easy to extend.

2. Strong empirical results: Experiments cover various models, datasets, and evaluation metrics. LIRE consistently outperforms or matches state-of-the-art methods. Scales well.

3. Well-written: The paper is clearly structured and easy to follow. Experiments are thorough. Limitations are discussed. Figures aid understanding.

**Weaknesses:**

While LIRE makes solid contributions, there are some limitations and weaknesses that could be addressed:

1. Narrow evaluation: Mainly tests dialogue and summarization. Could be extended to other LLM tasks.

2. Limited analysis: Does not perform ablation studies to isolate benefit of components. Hyperparameter sensitivity is unclear.

3. Lacks user studies: Human evaluations could provide more insight beyond automatic metrics.

Overall, while LIRE has impressive empirical performance, the theoretical analysis is limited. Exploring more advanced prompting strategies and scaling limits would strengthen the approach. But the paper makes excellent progress on an important problem.

**Questions:**

None

---

> ### Author Response · Authors · 2023-11-16
>
> We would like to thank the reviewer’s constructive comments and detailed review. We also thank the reviewer for pointing out some limitations.  We answer all questions below and update the manuscript accordingly.
>
> **Q$_1$: "Narrow evaluation: Mainly tests dialogue and summarization. Could be extended to other LLM tasks."**
>
> A$_1$: Thanks for pointing out the limitation of our evaluation. For other LLM tasks, we consider the controlled sentiment generation
> task. Specifically, given a prefix (20 tokens) of a movie review in the IMDb dataset, the models are asked to produce positive reviews. We use GPT-2-large as the base model and evaluate the comparing methods. As can be seen from the results list below, LIRE achieves the best RM score, which better illustrates the generalization ability of the proposed method. We will update the manuscript and add discussion about this task in the paper.
> | Eval. Metric | PPO   | DPO   | RRHF  | LIRE      |
> | ------------ | ----- | ----- | ----- | --------- |
> | PPL          | 15.69 | 14.71 | 14.38 | 17.11     |
> | RM           | 0.993 | 0.992 | 0.968 | **0.996** |
>
> **Q$_2$: "Limited analysis: Does not perform ablation studies to isolate benefit of components."**
>
> A$_2$: Our goal is to maximize the expectation of the final rewards and this is achieved by combining two components: the probability distribution and the reward scores, which are actually under the framework of policy gradient. Specifically, we reformulate the probability distribution among the multiple model generations through softmax normalization to incorporate information beyond a single pair, but a whole list of responses. So these two components are an integral whole, and can not be separated for optimization.
>
> **Q$_3$: "Hyperparameter sensitivity is unclear."**
>
> A$_3$: The main hyperparameter is the temperature T in Equation 4, and we demonstrate the performance under different T for the dialogue task, which is also included in Appendix Table 11. Larger T makes all the samples more uniformly weighted, while smaller T shifts the probability mass to the best sample (more like SFT training), therefore, T should be within a suitable range to boost performance.
> | T   | 1     | 2     | 5     | 10    | 20    |
> |-----|-------|-------|-------|-------|-------|
> | PPL | 13.76 | 12.61 | 13.67 | 12.65 | 13.55 |
> | RM  | -0.80 | -0.80 | -0.75 | -0.77 | -0.86 |
>
> **Q$_4$: "Lacks user studies: Human evaluations could provide more insight beyond automatic metrics."**
>
> A$_4$: Originally, we chose the evaluation metrics referring to other papers [1][2]. However, we do realize that this may not be a fair indicator. To ensure fairness for other methods and a more reliable comparison, aside from the automatic metrics, we also conduct human evaluation. The results are as follows. LIRE gains the highest win rate against human-written baselines as well as other comparing methods, which is in line with the metric results in general. We do realize the importance of conducting user study and we truly thank the reviewer for pointing out this crucial point. We update the manuscript accordingly to cover human evaluation results.
> |         vs.         |  ø   | PPO  | DPO  | PRO  | RRHF | LIRE |      |      |      |
> | :-----------------: | :--: | :--: | :--: | :--: | :--: | :--: | :--: | :--: | :--: |
> | human-written win % |  48  |  49  |  48  |  55  |  51  |  44  |      |      |      |
> |     LIRE win %      |  61  |  50  |  54  |  62  |  58  |  -   |      |      |      |
>
>
> [1].Yuan, Zheng, et al. "Rrhf: Rank responses to align language models with human feedback without tears." arXiv preprint arXiv:2304.05302 (2023).
>
> [2].Song, Feifan, et al. "Preference ranking optimization for human alignment." arXiv preprint arXiv:2306.17492 (2023).

---

### Official Review · Reviewer_6BAr · 2023-11-03

**Soundness:** 2 fair
**Presentation:** 2 fair
**Contribution:** 2 fair
**Rating:** 5
**Confidence:** 3

**Summary:**

The paper introduces LIRE, a new method that improves upon the standard Reinforcement Learning with Human Feedback for training Large Language Models. LIRE mitigates the generation of toxic content by LLMs by using a listwise approach to reward optimization, without relying on complex models or extensive hyperparameter tuning. The authors demonstrate that LIRE achieves stable performance and exceeds existing preference alignment methods, signifying a promising direction for creating safer LLMs.

**Strengths:**

1. The paper investigates the impact of a listwise loss function within the Reinforcement Learning with Human Feedback (RLHF) framework. The study elucidates the benefits of the listwise approach, particularly in terms of stability and efficacy.

2. By drawing connections between the proposed listwise loss method and Distributional Policy Optimization (DPO), the authors provide  theoretical insights. This comparison helps in positioning the proposed method within the broader landscape of reinforcement learning and policy optimization, enhancing the understanding of its advantages.

3. The experiments are comprehensive and results are promising. The authors conduct a various comparison between different methods with multiple evaluation metrics.

**Weaknesses:**

1. The main novelty of this paper lies in introduction of listwise loss and incorporating reward score into optimizations. For Listwise loss, DPO appendix also introduces how to extend from binary preference to multiple examples. The authors ignore this extension case and simply treat DPO as limited to binary preference. Without this comparison, the novelty of this paper is not clear.

2. The experimental results cannot clearly attribute performance improvements to the proposed components. More ablation study may help provide more insights, e.g., listwise loss may be the main factor or the strategy of introducing reward score. Another concern is in  metrics, which is not clearly discussed. The reward model is trained in optimization and may not be a good evaluation metric, especially this may not be fair for different model comparisons and  difficult for other readers to understand the level of performance. A side note: The presentation also needs improvement. The The abbreviation is suggested to be first introduced and then used, e.g., PPL.

**Questions:**

How to constrast LIRE and listwise version of DPO introduced in DPO paper appendix? What are the main conceptual differences and performance differences?

---

> ### Author Response · Authors · 2023-11-17
>
> We thank the reviewer for the detailed and constructive comments. We respond to the comments below and update the manuscript accordingly.
>
> **Q$_1$: "How to constrast LIRE and listwise version of DPO introduced in DPO paper appendix? What are the main conceptual differences and performance differences?"**
>
> A$_1$: Thanks for bringing up this constructive topic. The DPO objective is extended from binary preferences to multiple responses according to the Plackett-Luce model, and it resembles the top-k probability defined in [1], which gives a permutation probability distribution that relies on the position of a response in the permutation. However, in LIRE, we do not explicitly model an ordinal ranking, instead, the ranking (position) information is given by the reward scores. For an explicit and clear performance comparison, we also provide the results for summarization (TL;DR-3) and dialogue (HH-6) tasks. DPO performs slightly worse than LIRE on both tasks, which suggests that utilizing rewards explicitly helps align preference better. We update the manuscript accordingly to include the discussion of listwise DPO. Thanks for your suggestion!
>
>
> |        | **TL;DR-3** |          |          | **HH-6** |           |
> | ------ | ----------- | -------- | -------- | -------- | --------- |
> | Method | ROUGE-L     | RM-SUM   | RM-SUM*  | PPL      | RM        |
> | DPO    | 0.28        | 2.71     | 2.66     | 15.73    | -0.79     |
> | LIRE   | 0.23        | **2.88** | **3.00** | 12.45    | **-0.76** |
>
> **Q$_2$: "...cannot clearly attribute performance improvements to the proposed components. More ablation study may help provide more insights..."**
>
> A$_2$: Our goal is to maximize the expectation of the final rewards. This is achieved by combining two components: the probability distribution and the reward scores, which are actually under the framework of policy gradient. Specifically, we reformulate the probability distribution among the multiple model generations through softmax normalization to incorporate information beyond a single pair, but rather a whole list of responses. So these two components are an integral whole, and can not be separated for optimization.
>
> **Q$_3$: "...reward model is trained in optimization and may not be a good evaluation metric..."**
>
> A$_3$: Thanks for bringing this up. Initially, we chose the evaluation metrics referring to other papers [2][3]. However, we do realize that this may not be a fair indicator. To ensure fairness for other methods and a more reliable comparison, we also conduct human evaluation (47 feedback from volunteers, each consisting of 50 question samples) aside from the automatic metrics. The results are as follows. LIRE gains the highest win rate against both human-written baselines as well as other comparing methods, which is in line with the metric results in general. We do realize the importance of conducting human evaluation and we truly thank the reviewer for pointing out this crucial point. We update the manuscript accordingly to cover human evaluation results. Besides, we improve the presentation (e.g. PPL) in the paper and give a better discussion of the metrics.
> |         vs.         |  ø   | PPO  | DPO  | PRO  | RRHF | LIRE |      |      |      |
> | :-----------------: | :--: | :--: | :--: | :--: | :--: | :--: | :--: | :--: | :--: |
> | human-written win % |  48  |  49  |  48  |  55  |  51  |  44  |      |      |      |
> |     LIRE win %      |  61  |  50  |  54  |  62  |  58  |  -   |      |      |      |
>
> [1].Cao, Zhe, et al. "Learning to rank: from pairwise approach to listwise approach." Proceedings of the 24th international conference on Machine learning. 2007.
>
> [2].Yuan, Zheng, et al. "Rrhf: Rank responses to align language models with human feedback without tears." arXiv preprint arXiv:2304.05302 (2023).
>
> [3].Song, Feifan, et al. "Preference ranking optimization for human alignment." arXiv preprint arXiv:2306.17492 (2023).

---

### Official Review · Reviewer_Cmfo · 2023-11-05

**Soundness:** 2 fair
**Presentation:** 2 fair
**Contribution:** 2 fair
**Rating:** 5
**Confidence:** 4

**Summary:**

The paper proposes a new method for aligning large language models with human/ai feedback. As opposed to methods like PPO that involve a regularization objective, the authors propose LIRE which involves sampling n responses for each prompt and then maximizing the expected reward on the sampled responses.

The proposed objective is differentiable, as it only involves a softmax over the candidate generations. The authors claim that since the method involves sampling from the aligned policy model the method does not require regularizing the policy to the anchor model.

**Strengths:**

* Alignment is an important problem in AI and newer methods for alignment are welcome since they can potentially engender discussion and help the community progress.
* Current methods rely on regularization objectives [PPO, SLiC] to ensure the aligned policy does not deviate from the anchor policy model, the problem of policy divergence and reward over-optimization is an important one. In as much contributions that improve robustness of alignment techniques are welcome.
* The authors benchmark their approach on two real world datasets and provide qualitative results in the supplemental sections suggesting thoroughness in terms of evaluation,

**Weaknesses:**

* The paper makes some very strong conjectures without substantial backing of their claims. one such instances are
- Section 5.5 The authors claim that their objective implicitly includes the SFT objective? This is a very strong claim, and I do not believe this is the case. Unless the authors can demonstrate this mathematically I would suggest the authors tone down their narrative.

* The authors claim that adding SFT loss would prevent the model from reward over-optimization. This is incorrect! Unless the alignment strategy also includes the pretraining loss, adding just the SFT loss would lead to the model collapse. This is exactly the reason regularization objectives like KL-div are including in alignment objectives, as they make the aligned policy close to the SFT policy, which preserves knowledge from the pretraining steps as well.

* Another line of work that is similar and involves sampling of multiple responses is Sequence Likelihood Calibration SLiC (Zhao et al, 2022) which the author have not considered or benchmarked against, a major omission in my opinion.

* Finally the paper does NOT consider "list-wise" approaches at all. In the case of listwise approaches, loss functions are designed in an ordinal fashion that ranks candidates in a list order. In this paper however the authors sample M generations, compute their likelihoods and take an expectation over the reward signal under the M generations, using a softmax. This is an "absolute" estimation of reward and claiming this to be "listwise" is gravely misleading.

**Questions:**

- Can the authors provide evidence in terms of divergence metrics/regularization metrics between a policy tuned with LIRE and the anchor policy to prove that the LIRE objective ensures closeness to the supervised policy?

---

> ### Author Response · Authors · 2023-11-17
>
> We thank the reviewer for the very detailed and constructive comments. Below are our responses to all the questions and comments, and we have updated the manuscript accordingly. We hope that the responses are reasonable and satisfactory enough to address the reviewer’s concerns and we welcome further discussion.
>
> **Q$_1$: "...claim that their objective implicitly includes the SFT objective?..."**
>
> A$_1$: According to the derivative of the LIRE objective in Equation 6, the gradient of each sampled response is weighted according to the reward scores. For queries that include human-annotated responses in the candidate list,  LIRE includes the human-annotation during loss calculation, and thus the human-preferred response also contributes to the gradient decent. This can be perceived as an implicit SFT loss component in a sense. Additionally, we can enforce an explicit SFT loss to better adhere to human preferences. We will modify the relevant script and tone down the narrative to make the paper more rigorously presented. Thanks for the advice!
>
> **Q$_2$: "...adding just the SFT loss would lead to the model collapse. This is exactly the reason regularization objectives like KL-div are including in alignment objectives..."**
>
> A$_2$: Thanks for pointing out the different effects of SFT loss and KL divergence and we really appreciate that.
>  We conduct experiments adding KL divergence to the LIRE objective and compare the averaged rewards under different KL penalties. We report averaged sequence-level KL and the corresponding averaged reward scores. The results demonstrate that just adding SFT loss will, as the reviewer suggested, not help preserve knowledge from the pretraining steps. Increasing the level of KL penalty helps mitigate the divergence between the training policy and the anchor policy, at the sacrifice of some reward losses. We update the manuscript in Section 5.5 accordingly and will cover the discussion of KL divergences.
>
> |                                 | LIRE  | LIRE+0.01*KL | LIRE+0.05*KL | LIRE+0.1*KL | LIRE+0.01*SFT |
> | ------------------------------- | ----- | ----------- | ----------- | ---------- | ------------ |
> | KL($\pi_{\theta}\|\|\pi_{ref}$) | 9.75  | 9.07       | 8.74        | 8.33      | 11.73       |
> | Reward score                    | -0.847 | -0.849       | -0.870       | -1.084     | -0.817        |
>
> **Q$_3$: "Can the authors provide evidence in terms of divergence metrics/regularization metrics between a policy tuned with LIRE and the anchor policy to prove that the LIRE objective.."**
>
> A$_3$: As suggested in the above table, LIRE has a KL divergence metric of 9.75, which is slightly larger than 9.07 (0.01KL penalty).  Adding just SFT loss to LIRE actually leads to a larger divergence of 11.73, despite achieving the best reward score of -0.817, which validates the reviewer's point made in **Q$_2$**. We also observe a tendency for reward decrease when increasing the level of KL penalty, and LIRE objective
> achieves a good balance between reward score and KL divergence against the anchor policy. We thank the reviewer for pointing out this constructive topic and will include more discussion on this in the manuscript.

---

> ### Author Response · Authors · 2023-11-17
>
> **Q$_4$: "...Another line of work that is similar and involves sampling of multiple responses is Sequence Likelihood Calibration SLiC..."**
>
> A$_4$: Thanks for pointing out this omission. We try to compare with the SLiC objective, specifically, we choose the rank calibration loss and cross-entropy regularization loss for the pairwise human feedback as introduced in [1] for the dialogue and summarization task. The comparison results are as follows. SLiC achieves good performance on TL;DR while performing worse on the HH task, and LIRE still achieves better performance. We will include the discussion of SLiC[2] in the paper for a more complete comparison.
>
> |         | TL;DR Summarization |          | HH Dialogue |           |
> | ------- | ------------------- | -------- | ----------- | --------- |
> | Methods | ROUGE-L             | RM-SUM   | PPL         | RM        |
> | SLiC-HF | 0.22                | 2.57     | 14.09       | -1.12     |
> | DPO     | 0.29                | 2.14     | 16.04       | -0.87     |
> | LIRE    | 0.22                | **2.76** | 12.15       | **-0.85** |
>
> **Q$_5$: "Finally the paper does NOT consider "list-wise" approaches at all."**
>
> A$_5$: This is a good point. Essentially, LIRE does not rely on an ordinal ranking, instead, the ranking (position) information is given by the reward scores. This is different from the top-k probability defined in ListNet[3], which gives a permutation probability distribution that relies on the position of a response in the permutation. In LIRE, the sampled model generations are normalized with softmax, which means information of the whole list is incorporated into a single response in the list, with different reward scores weighing each response differently in loss calculation according to Equation 6. The reward scores bring a more quantified differentiation for each sample compared to ordinal ranking. This is different from standard "listwise" paradigms and we are sorry for the confusion. We have updated the manuscript accordingly to give a clearer illustration of this idea.
>
>
> [1].Zhao, Yao, et al. "Slic-hf: Sequence likelihood calibration with human feedback." *arXiv preprint arXiv:2305.10425* (2023).
>
> [2].Zhao, Yao, et al. "Calibrating sequence likelihood improves conditional language generation." *arXiv preprint arXiv:2210.00045* (2022).
>
> [3].Cao, Zhe, et al. "Learning to rank: from pairwise approach to listwise approach." *Proceedings of the 24th international conference on Machine learning*. 2007.

---

### Official Review · Reviewer_Z7nt · 2023-11-05

**Soundness:** 2 fair
**Presentation:** 2 fair
**Contribution:** 3 good
**Rating:** 5
**Confidence:** 3

**Summary:**

1. Motivated by moving from a pairwise loss between different generations from a language model to a list-wise approach to directly model the list of (ranked) generations, the paper proposes LIRE: Listwise Reward Enhancement for Preference Alignment.
2. Concretely, the paper defines a distribution over a list of candidate generations given a prompt, with the probability of each of the instance of the list is proportional to the the probability of the generated sequence under the model (with temperature smoothing)
3. The authors then define a loss using the aforementioned distribution, weighing each sample in the list with its associated reward from a reward model.
4. The authors also propose a self-refinement approach for iteratively improving the base policy.
5. The experiments demonstrate the proposed method obtains improvements over baseline methods for both RM score, perplexity score as well as an LLM prompted evaluation metric for both dialogue generation on Helpful and Harmless dataset, as well as on TL;DR abstract summarization.
6. Finally the authors also demonstrate that their proposed method continually improves the reward score and reduces variance with increased compute for the self-refinement stage.

**Strengths:**

1. The proposed methodology for moving from a pairwise approach to a list-wise approach for alignment is well motivated, especially with the ranking information between different LLM model generations becoming increasingly available.
2. The connection drawn between the proposed method and other direct policy improvement methods like DPO (Page 5) is quite informative in providing a different perspective for the gradient update step.
3. The improvements from LIRE are quite impressive, especially on the MT-Bench. The proposed method does out-perform the other SoTA approaches.

**Weaknesses:**

1. The authors claim that their proposed approach does not require a KL constraint. However, as presented in [1], without a KL constraint, alignment training would lead to a distributional collapse. In my opinion, by training on generations from other (somewhat aligned) LLMs, the authors implicitly leverage the KL constraint, and hence this claim seems a bit strong.

2. The generative distribution defined by the authors is very confusing (Equation 4). As per my understanding, this should be similar in spirit to the top k probability distribution, defined in Definition 7 in [2]. However, that does not seem to be the case. Specifically,

2.1 \pi_{theta} seems to refer to the log-prob distribution, but in other places, is also referred to as the sampling distribution. This makes it very confusing to understand exactly what this quantity is supposed to model.

2.2 Equation 3 seems to model generations as P_{\pi_{\theta}} (y^{i}_{j,k} | x^{i}) , but for autoregressive models that the authors study, the probability should be modelled as P_{\pi_{theta}}(y^{i}_{j,k} | x^{i}, y^{i}_{j, <k}), which in turn renders it intractable to compute

2.3 The reward itself is computed from a model trained on pairwise comparison data. This seems somewhat opposed to the core problem that the authors are trying to solve: of moving from pair-wise to list-wise approach for modelling alignment interactions.

3. In Algorithm 1, sampling from \pi_{\theta} seems intractable (or at least would require setting up a Monte-Carlo chain which would be computationally pretty expensive). The details seem to be missing for this crucial step.

4. The metrics used for evaluation, particularly reward model scores and perplexity under a GPT2 model seem underspecified to be able to compare if a model / approach is better aligned compared to another. Specifically, having a much higher reward under an estimated reward model can be because of improved alignment or because of spurious correlations in the reward modelling dataset which the RM might pick up on. Likewise, perplexity under GPT2 medium model estimates (up to a certain degree) to a certain degree how close the final model is compared to the reference policy (here the GPT2 medium model), which may not directly imply a better aligned model. Having additional win rate metrics, especially comparing two algorithms directly (eg: LIRE vs PPO) might be better to answer the question on which model is better

Overall, the proposed method does seem to improve over baseline approaches, which is quite valuable. Additionally, the problem of studying list-wise ranking methodologies for alignment training is a valid one. However, the ambiguities around the motivation for the approach does raise some questions.

[1] Korbak, Tomasz, Ethan Perez, and Christopher L. Buckley. "RL with KL penalties is better viewed as Bayesian inference." arXiv preprint arXiv:2205.11275 (2022).
[2] Cao, Zhe, et al. "Learning to rank: from pairwise approach to listwise approach." Proceedings of the 24th international conference on Machine learning. 2007.

**Questions:**

Questions:

1. I am very unclear on how the generative sampling is done under the proposed distribution over top-k generations. (Line 2 in the Algorithm 1.) I would be grateful for any clarifications on the same.
2. Would it be possible to differentiate between the policy distribution and the log-probability under the model for a generation. The intermixing of both causes a fair bit of confusion in understanding the paper.
3. In Table 1, it seems like the PPO model achieves worse reward compared to a baseline Alpaca-7B model, which seems very counter-intuitive, given that that is what the PPO model should be optimizing for. Would it be possible to provide any intuition on why this might be the case ?
4. How many samples from the test split are considered for the GPT-4 evaluation ?
5. In Figure 3, it seems like after LIRE, there are a fair number of samples for which the RM scores actually reduced (below the diagonal and after RM = 0 on the X axis). Is there any intuition on why that might have happened ?

---

> ### Author Response · Authors · 2023-11-17
>
> We appreciate the thorough and constructive feedback from the reviewer. Our responses to all questions and comments are provided below. We hope that the responses are reasonable and satisfactory enough to address the reviewer’s concerns and we welcome further discussion.
>
> **Q$_1$: "...how the generative sampling is done under the proposed distribution over top-k generations."**
>
> $A_1$: Line 2 in Algorithm 1 is the offline dataset generation process where we gather $\textbf{offline}$ training data. In the experiment, the preference data was originally gathered from both human preferences as well as from Alpaca-7B using diverse beam search (when $e$=1); for $e$>1, we can additionally leverage the self-enhancement strategy proposed in Algorithm 1 and use the trained policy after $I$ epoch to generate new offline data to refresh the previous candidate pool (this practice is also adopted in ResT[1]).  During training, the generative distribution defined in Equation 4 actually takes a softmax over the $m$ sampled model generations. This is different from the top-k probability defined in ListNet[2], which gives a permutation probability distribution that relies on the $\textbf{position}$ of a response in the permutation. In LIRE, the sampled model generations are normalized with softmax, with different reward scores weighing each response differently in loss calculation according to Equation 6, and no explicit ranking is required during this process.
>
> **Q$_2$: "Would it be possible to differentiate between the policy distribution and the log-probability under the model for a generation“**
>
> $A_2$: We are sorry for the confusion and have modified the manuscript. Now $\pi_{\theta}$ in Equation 4 refers to the model sampling distribution which aligns with other papers. We also fixed Equation 3 for autoregressive model generation $P_{\pi_{\theta}}(y_{j,k}^{(i)}\|x^{(i)},y^{(i)}_{j,<k})$. Thanks for the reviewer's detailed review!
>
> **Q$_3$: "...it seems like the PPO model achieves worse reward compared to a baseline Alpaca-7B model, which seems very counter-intuitive..."**
>
> A$_3$: Alpaca-7B is finetuned using the Alpaca dataset [3],  in which the root verb of "summarize" makes up a very small fraction of the whole dataset, while other root verbs of "write", "suggest", "explain", etc., consist a large fraction, and these topics share more similarity of the HH dataset. This may account for Alpaca being a strong performer for questions in HH, however, in the summarization task, PPO performs significantly better than Alpaca-7B.
>
> **Q$_4$: "How many samples from the test split are considered for the GPT-4 evaluation ?"**
>
> A$_4$: For the GPT-4 evaluation, we use 100 samples from the test split for each method for comparison. Since it is verified in [4][5], that GPT-4 judgment has a high agreement with human evaluation, we think 100 questions should be reasonable to obtain a GPT-4-based estimation that reflects human preference.  Besides, the MT and Vicuna Bench both consist of 80 questions for evaluation using GPT-3.5 and GPT-4. For the above reasons, we chose 100 questions for GPT-4 evaluation.
>
> **Q$_5$: "It seems like after LIRE, there are a fair number of samples for which the RM scores actually reduced..."**
>
> A$_5$: This is quite an interesting point. We visualize all the instance-level RM variation for each comparing method and find that all the methods display a fair large amount of reward reduction after training, which seems to be a common issue for different methods. However, compared to other methods, LIRE has a smaller ratio of instance-level reward drop across the test samples and better RM improvement on average. We list the percentage of the samples with declining scores in the whole test set for each method and LIRE has the smallest ratio of 27% of score decrease. We speculate that it may be due to factors such as bias during the data sampling process, which will be left as our future work. Thanks for the reviewer's question and we have added more visualization results in Appendix A.9.
>
> |                 | PPO  | DPO  | PRO  | RRHF | LIRE   |
> | --------------- | ---- | ---- | ---- | ---- | ------ |
> | Decrease rate % | 57   | 41   | 46   | 52   | **27** |

---

> ### Author Response · Authors · 2023-11-17
>
> **Q$_6$: "The authors claim that their proposed approach does not require a KL constraint..."**
>
> A$_6$: This is a good point. According to Equation 6, the gradient of each sampled response is weighted according to the reward scores. For queries that include human-annotated responses in the candidate list, LIRE naturally includes the human-annotation contributing to the gradient descent. This can be perceived as an implicit SFT loss. Additionally, we can enforce an explicit SFT loss to better adhere to human preferences. According to [6], Cross entropy (the same as SFT loss under the MLE framework) and KL divergence regularization perform similarly and prevent models from deviating significantly from their fine-tuned objective. However, we do realize that without a KL constraint, alignment training would lead to a distributional collapse. We have modified the presentation and thanks for bringing this out.
>
> **Q$_7$: "The reward itself is computed from a model trained on pairwise comparison data. This seems somewhat opposed to the core problem.."**
>
> A$_7$: The reward model is trained on pairwise preference, and the comparison between pairs of data is transitive to the whole list of data, given any input data. This partial order relationship modeled by conditional probabilities can be used to score an entire sequence comparably, which means this pairwise preference training for reward model is actually a surrogate for achieving list-wise comparison. So this is in line with our list-wise approach.
>
> **Q$_8$: "Having additional win rate metrics, especially comparing two algorithms directly..."**
>
> A$_8$: Thanks for pointing out a better way for performance comparison. To ensure fairness for other methods and a more reliable comparison, aside from the automatic metrics, we also conduct human evaluation. The results are as follows. LIRE gains the highest win rate against human-written baselines as well as other comparing methods, which is in line with the metric results in general. We do realize the importance of conducting user study and we truly thank the reviewer for pointing out this crucial point. We update the manuscript accordingly to cover human evaluation results.
>
> |         vs.         |  ø   | PPO  | DPO  | PRO  | RRHF | LIRE |      |      |      |
> | :-----------------: | :--: | :--: | :--: | :--: | :--: | :--: | :--: | :--: | :--: |
> | human-written win % |  48  |  49  |  48  |  55  |  51  |  44  |      |      |      |
> |     LIRE win %      |  61  |  50  |  54  |  62  |  58  |  -   |      |      |      |
>
> [1] Gulcehre, Caglar, et al. "Reinforced self-training (rest) for language modeling." arXiv preprint arXiv:2308.08998 (2023).
>
> [2] Cao, Zhe, et al. "Learning to rank: from pairwise approach to listwise approach." Proceedings of the 24th international conference on Machine learning. 2007.
>
> [3] https://github.com/tatsu-lab/stanford_alpaca
>
> [4].Song, Feifan, et al. "Preference ranking optimization for human alignment." arXiv preprint arXiv:2306.17492 (2023).
>
> [5]. Rafailov, Rafael, et al. "Direct preference optimization: Your language model is secretly a reward model." *arXiv preprint arXiv:2305.18290* (2023).
>
> [6].Zhao, Yao, et al. "Calibrating sequence likelihood improves conditional language generation." *arXiv preprint arXiv:2210.00045* (2022).

---

### Author Response · Authors · 2023-11-17

We’d like to thank all the reviewers for providing high-quality and constructive feedback. We try to respond to all the questions and have updated the manuscript accordingly.

To summarize:

1. Conducted user study for a fair comparison among different methods.
2. Added many related work suggestions.
3. Modified the expression in the manuscript to improve the presentation.

Please see below for reviewer-specific discussions.

We would appreciate it if any necessary change is made to the review if the responses are reasonable and satisfactory enough to address the reviewers' concerns. Please let us know if there are unresolved questions and we welcome further feedback.

---

### Meta-Review · Area_Chair_76Qk · 2023-12-05

**Metareview:**

This paper proposes list-wise reward enhancement for preference alignment. The gist of the method is as follows. For a set of N alignment prompts, the base model is rolled out to generate a list of m responses. Then the objective function given in Eq. (5) is to make the expected reward higher by making the higher reward responses more likely under the model. The initial experiments show that the method is promising in improving alignment (win-rate against base policy). While the initial results are quite interesting and the proposed method seems promising there are a few major concerns that lead to the decision o reject at this point:

1. The KL divergence between the aligned policy and the base policy should be characterized as a proxy for how much the policy has drifted from the base policy. As such, it is customary to explore the reward (win-rate) vs KL tradeoffs. Hence the comparison with DPO, PPO and other methods is apples-to-oranges at this time.

2. A few other simple additional baselines should be compared against. (a) inference-time best-of-m; (b) SFT+KL-regularization on the best-of-m response from the pre-computed responses. This is to make sure that the proposed complicated method is indeed giving non-trivial gains against these much simpler baselines.

3. The method needs to be better understood, specifically in contrast to DPO. Currently, it is not clear how to reason about this proposal as compared with other alignment techniques, all of which intuitively make good responses more likely and bad ones less likely.

We hope that the authors can take the extensive feedback from the reviewers into account for a future submission.

**Justification For Why Not Higher Score:**

The justification is presented in three bullet points:

1. The KL divergence between the aligned policy and the base policy should be characterized as a proxy for how much the policy has drifted from the base policy. As such, it is customary to explore the reward (win-rate) vs KL tradeoffs. Hence the comparison with DPO, PPO and other methods is apples-to-oranges at this time.

2. A few other simple additional baselines should be compared against. (a) inference-time best-of-m; (b) SFT+KL-regularization on the best-of-m response from the pre-computed responses. This is to make sure that the proposed complicated method is indeed giving non-trivial gains against these much simpler baselines.

3. The method needs to be better understood, specifically in contrast to DPO. Currently, it is not clear how to reason about this proposal as compared with other alignment techniques, all of which intuitively make good responses more likely and bad ones less likely.

**Justification For Why Not Lower Score:**

The method is appealing in terms of implementation and the initial experiments seem promising.

---

### Decision · Program_Chairs · 2024-01-16

Reject